**A global hotspot for dissolved organic carbon in hypermaritime watersheds of coastal**
**British Columbia.**
Allison A. Oliver[1,2], Suzanne E. Tank[1,2], Ian Giesbrecht[2,7], Maartje C. Korver[2], William C.
Floyd[3,4,2], Paul Sanborn[5,2], Chuck Bulmer[6], Ken P. Lertzman[7,2]
[1]University of Alberta, Department of Biological Sciences, CW 405, Biological Sciences Bldg.,
University of Alberta, Edmonton, Alberta, T6G 2E9, Canada
[2]Hakai Institute, Tula Foundation, Box 309, Heriot Bay, British Columbia, V0P 1H0, Canada
[3]Ministry of Forests, Lands and Natural Resource Operations, 2100 Labieux Rd, Nanaimo, BC,
V9T 6E9, Canada
[4]Vancouver Island University, 900 Fifth Street, Nanaimo, BC, V9R 5S5, Canada
[5]Ecosystem Science and Management Program, University of Northern British Columbia, 3333
University Way, Prince George, BC, V2N 4Z9, Canada
[6]BC Ministry of Forests Lands and Natural Resource Operations, 3401 Reservoir Rd, Vernon,
BC, V1B 2C7, Canada
[7]School of Resource and Environmental Management, Simon Fraser University, TASC 1- Room
8405, 8888 University Drive, Burnaby, BC, V5A 1S6, Canada
Corresponding author: aaoliver@ualberta.ca
**Abstract**

The perhumid region of the coastal temperate rainforest (CTR) of Pacific North America

is one of the wettest places on Earth and contains numerous small catchments that discharge
freshwater and high concentrations of dissolved organic carbon (DOC) directly to the coastal
ocean. However, empirical data on the flux and composition of DOC exported from these
watersheds is scarce. We established monitoring stations at the outlets of seven catchments on
Calvert and Hecate Islands, British Columbia, which represent the rain dominated hypermaritime
region of the perhumid CTR. Over several years, we measured stream discharge, stream water
DOC concentration, and stream water dissolved organic matter (DOM) composition. Discharge
and DOC concentrations were used to calculate DOC fluxes and yields, and DOM composition
was characterized using absorbance and fluorescence spectroscopy with parallel factor analysis
(PARAFAC). The areal estimate of annual DOC yield in water year 2015 was 33.3 Mg C km$^{-2}$
yr$^{-1}$, with individual watersheds ranging from an average of 24.1-37.7 Mg C km$^{-2}$ yr$^{-1}$. This
represents some of the highest DOC yields to be measured at the coastal margin. We observed
seasonality in the quantity and composition of exports, with the majority of DOC export
occurring during the extended wet period (September-April). Stream flow from catchments
reacted quickly to rain inputs, resulting in rapid export of relatively fresh, highly terrestrial-like
DOM. DOC concentration and measures of DOM composition were related to stream discharge
and stream temperature, and correlated with watershed attributes, including the extent of lakes
and wetlands, and thickness of organic and mineral soil horizons. Our discovery of high DOC
yields from these small catchments in the CTR is especially compelling as they deliver relatively
fresh, highly terrestrial organic matter directly to the coastal ocean. Hypermaritime landscapes
are common on the British Columbia coast, suggesting that this coastal margin may play an
important role in the global processing of carbon and in linking terrestrial carbon to marine
ecosystems.

## 1. Introduction

Freshwater aquatic ecosystems process and transport a significant amount of carbon
(Cole et al., 2007; Aufdenkampe et al., 2011; Dai et al., 2012). Globally, riverine export is
estimated to deliver around 0.9 Pg C yr$^{-1}$ from land to the coastal ocean (Cole et al., 2007), with
typically >50% quantified as dissolved organic carbon (DOC)(Meybeck, 1982; Ludwig et al.,
1996; Alvarez-Cobelas et al., 2012; Mayorga et al., 2010). Rivers draining coastal watersheds
serve as conduits of DOC from terrestrial and freshwater sources to marine environments
(Mulholland and Watts, 1982; Bauer et al., 2013; McClelland et al., 2014) and can have
important implications for coastal carbon cycling, biogeochemical interactions, ecosystem
productivity, and food webs (Hopkinson et al., 1998; Tallis, 2009; Tank et al., 2012; Regnier et
al., 2013). In addition, because the transfer of water and organic matter from watersheds to the
coastal ocean represents an important pathway for carbon cycling and ecological subsidies
between ecosystems, better understanding of these linkages is needed for constraining
predictions of ecosystem productivity in response to perturbations such as climate change. In
regions where empirical data are currently scarce, quantifying land-to-ocean DOC export is
therefore a priority for improving the accuracy of watershed and coastal carbon models (Bauer et
al., 2013).
While quantifying DOC flux within and across systems is required for understanding the
magnitude of carbon exchange, the composition of DOC (as dissolved organic matter, or DOM)
is also important for determining the ecological significance of carbon exported from coastal
watersheds. The aquatic DOM pool is a complex mixture that reflects both source material and
processing along the watershed terrestrial-aquatic continuum, and as a result can show
significant spatial and temporal variation (Hudson et al., 2007; Graeber et al., 2012; Wallin et al.,
2015). Both DOC concentration and DOM composition can serve as indicators of watershed
characteristics (Koehler et al., 2009), hydrologic flow paths (Johnson et al., 2011; Helton et al.,
2015), and watershed biogeochemical processes (Emili and Price, 2013). DOM composition can
also influence its role in downstream processing and ecological function, such as susceptibility to
biological (Judd et al., 2006) and physiochemical interactions (Yamashita and Jaffé, 2008).

The coastal temperate rainforests (CTR) of Pacific North America extend from the Gulf

of Alaska, through British Columbia, to Northern California and span a wide range of
precipitation and climate regimes. Within this rainforest region, the "perhumid" zone has cool
summers and summer precipitation is common (>10% of annual precipitation) (Alaback,
1996)(Fig. 1). The perhumid CTR extends from southeast Alaska through the outer coast of
central British Columbia and contains forests and soils that have accumulated large amounts of
organic carbon above and below ground (Leighty et al., 2006; Gorham et al., 2012). Due to high
amounts of precipitation and close proximity to the coast, this area represents a potential hotspot
for the transport and metabolism of carbon across the land-to-ocean continuum, and quantifying
these fluxes is pertinent for understanding global carbon cycling.

Within the large perhumid CTR, there is substantial spatial variation in climate and

landscape characteristics that create uncertainty about carbon cycling and pattern. In Alaska, for
example, riverine DOC concentrations vary with wetland cover (D'Amore et al. 2015a) and
glacial cover (Fellman et al. 2014). Previous studies have shown that streams in southeast Alaska
can contain high DOC concentrations (Fellman et al., 2009a; D'Amore et al., 2015a) and
produce high DOC yields (D'Amore et al., 2015b; D'Amore et al., 2016, Stackpoole et al.,
2016), but no known field estimates have been generated for the perhumid CTR of British
Columbia, an area of approximately 97,824 km$^2$ (adapted from Wolf et al., 1995). Within the
perhumid CTR of British Columbia, terrestrial ecologists have defined a large (29,935 km$^2$)
*hypermaritime* sub-region where rainfall dominates over snow, seasonality is moderated by the
ocean, and wetlands are extensive (Pojar et al., 1991; area estimated using British Columbia
Biogeoclimatic Ecosystem Classification Subzone/Variant mapping Version 10, August 31,
2016, available at: https://catalogue.data.gov.bc.ca/dataset/f358a53b-ffde-4830-a325-
a5a03ff672c3). Previous work in the hypermaritime CTR showed that DOC concentrations are
high in small streams and tend to increase during rain events (Gibson et al., 2000; Fitzgerald et
al., 2003; Emili and Price, 2013). Taken together, these conditions should be expected to
generate high yields and fluxes of DOC from hypermaritime watersheds to the coastal ocean.

The objectives of this study were to provide the first field-based estimates of DOC

exports from watersheds in the extensive hypermaritime region of British Columbia's perhumid
CTR, to describe the temporal and spatial dynamics of exported DOC concentration and DOM
composition, and to identify relationships between DOC concentration, DOM composition, and
watershed characteristics.
**2. Methods**
**2.1 Study Sites**

Study sites are located on northern Calvert Island and adjacent Hecate Island on the

central coast of British Columbia, Canada (Lat 51.650, Long -128.035; Fig. 1). Average annual
precipitation and air temperature at sea level from 1981-2010 was 3356 mm yr$^{-1}$ and 8.4 °C
(average annual min= 0.9°C, average annual max= 17.9°C) (available online at
http://www.climatewna.com/; Wang et al., 2012), with precipitation dominated by rain, and
winter snowpack persisting only at higher elevations. Sites are located within the hypermaritime
region of the CTR on the outer coast of British Columbia. Soils overlying the granodiorite
bedrock (Roddick, 1996) are usually < 1 m thick, and have formed in sandy colluvium and
patchy morainal deposits, with limited areas of coarse glacial outwash. Chemical weathering and
organic matter accumulation in the cool, moist climate have produced soils dominated by
Podzols and Folic Histosols, with Hemists up to 2 m thick in depressional sites (IUSS Working
Group WRB, 2015). The landscape is comprised of a mosaic of ecosystem types, including
exposed bedrock, extensive wetlands, bog forests and woodlands, with organic rich soils (Green,
2014; Thompson et al., 2016). Forest stands are generally short with open canopies reflecting the
lower productivity of the hypermaritime forests compared to the rest of the perhumid CTR
(Banner et al., 2005). Dominant trees are western redcedar, yellow-cedar, shore pine and western
hemlock with composition varying across topographic and edaphic gradients. Widespread
understory plants include bryophytes, salal, deer fern, and tufted clubrush. Wetland plants are
locally abundant including diverse *Sphagnum* mosses and sedges. Although the watersheds have
no history of mining or industrial logging, archaeological evidence suggests that humans have
occupied this landscape for at least 13,000 years (McLaren et al., 2014). This occupation has had
a local effect on forest productivity near habitation sites (Trant et al., 2016) and on fire regimes
(Hoffman et al., 2016). We selected seven watersheds with streams draining directly into the
ocean (Fig. 1). These numbered watersheds (626, 693, 703, 708, 819 844, and 1015) range in
size (3.2 to 12.8 km$^2$) and topography (maximum elevation 160 m to 1012 m), are variably
affected by lakes (0.3 – 9.1% lake coverage), and – as is characteristic of the perhumid CTR–
have a high degree of wetland coverage (24– 50%) (Table 1).
**2.2 Soils and watershed characteristics**
Watersheds and streams were delineated using a 3 m resolution digital elevation model
(DEM) derived from airborne laser scanning (LiDAR) and flow accumulation analysis using
geographic information systems (GIS) to summarize watershed characteristics for each
watershed polygon and for all watersheds combined (Gonzalez Arriola et al., 2015; Table 1).
Topographic measures were estimated from the DEM, and lake and wetland cover estimated
from Province of British Columbia Terrestrial Ecosystem Mapping (TEM) (Green, 2014), and
soil material thickness estimated from unpublished digital soil maps (Supplemental S1). We
recorded thickness of organic soil material, thickness of mineral soil material, and total soil depth
to bedrock at a total of 353 field sites. Mineral soil horizons have $\leq$ 17% organic C, while
organic soil horizons have > 17% organic C, per the Canadian System of Soil Classification (Soil
Classification Working Group, 1998). In addition to field-sampled sites, 40 sites with exposed
bedrock (0 cm soil depth) were located using aerial photography. Soil thicknesses were
combined with a suite of topographic, vegetation, and remote sensing (LiDAR and RapidEye
satellite imagery) data for each sampling point and used to train a random forest model
(randomForest package in R; Liaw and Wiener, 2002) that was used to predict soil depth values.
Soil material thicknesses were then averaged for each watershed (Table 1). For additional details
on field site selection and methods used for predictions of soil thickness, see Supplemental S1.1.
**2.3 Sample Collection and Analysis**
From May 2013 to July 2016, we collected stream water grab samples from each
watershed stream outlet every 2-3 weeks ($n_{total}$= 402), with less frequent sampling (~ monthly)
during winter (Fig. 1). All samples were filtered in the field (Millipore Millex-HP Hydrophilic
PES 0.45µm) and kept in the dark, on ice until analysis. DOC samples were filtered into 60 mL
amber glass bottles and preserved with 7.5M $H_3PO_4$.  Fe samples were filtered into 125 mL
HDPE bottles and preserved with 8M $HNO_3$.  DOC and Fe samples were analyzed at the BC
Ministry of the Environment Technical Services Laboratory (Victoria, BC, Canada). DOC
concentrations were determined on a TOC analyzer (Aurora 1030; OI-Analytical) using wet
chemical oxidation with persulfate followed by infrared detection of $CO_2$.  Fe concentrations
were determined on a dual-view ICP-OES spectrophotometer (Prodigy; Teledyne Leeman Labs)
using a Seaspray pneumatic nebulizer.

In May 2014, we began collecting stream samples for stable isotopic composition of $\delta^{13}C$

in DOC ($\delta^{13}C$-DOC; n= 173) and optical characterization of DOM using absorbance
spectroscopy (n= 259). Beginning in January 2016, we also analyzed samples using fluorescence
spectroscopy (see section 2.6). Samples collected for $\delta^{13}C$-DOC were filtered into 40 mL EPA
glass vials and preserved with $H_3PO_4$. $\delta^{13}C$-DOC samples were analyzed at GG Hatch Stable
Isotope Laboratory (Ottawa, ON, Canada) using high temperature combustion (TIC-TOC
Combustion Analyser Model 1030; OI Analytical) coupled to a continuous flow isotope ratio
mass spectrometry (Finnigan Mat DeltaPlusXP; Thermo Fischer Scientific)(Lalonde et al. 2014).
Samples analyzed for optical characterization using absorbance and fluorescence were filtered
into 125 mL amber HDPE bottles and analyzed at the Hakai Institute (Calvert Island, BC,
Canada) within 24 hours of collection.
**2.4 Hydrology: Precipitation and Stream Discharge**

We measured precipitation using a TB4-L tipping bucket rain gauge with a 0.2 mm

resolution (Campbell Scientific Ltd.) located in watershed 708 (elevation= 16 m a.s.l). The rain
gauge was calibrated twice per year using a Field Calibration Device, model 653 (HYQUEST
Solutions Ltd).
We determined continuous stream discharge for each watershed by developing stage
discharge rating curves at fixed hydrometric stations situated in close proximity to each stream
outlet.  Sites were located above tidewater influence and were selected based on favourable
conditions (i.e., channel stability and stable hydraulic conditions) for the installation and
operation of pressure transducers to measure stream stage. From August 2014 to May 2016 (21
months), we measured stage every 5 minutes using an OTT PLS –L (OTT Hydromet, Colorado,
USA) pressure transducer (0-4 m range SDI-12) connected to a CR1000 (Campbell Scientific,
Edmonton, Canada) data logger. Stream discharge was measured over various intervals using
either the velocity area method (for flows $< 0.5$ m$^3$s$^{-1}$; ISO Standard 9196:1992, ISO Standard
748:2007) or salt dilution (for flows $> 0.5$ m$^3$s$^{-1}$; Moore, 2005). Rating curves were developed
using the relationship between stream stage height and stream discharge (Supplemental S2).
**2.5 DOC flux**
From October 1, 2014 to April 30, 2016, we estimated DOC flux for each watershed
using measured DOC concentrations (n= 224) and continuous discharge recorded at 15-minute
intervals. The watersheds in this region respond rapidly to rain inputs and as a result DOC
concentrations are highly variable. To address this variability, routine DOC concentration data
(as described in section 2.2) were supplemented with additional grab samples (n= 21) collected
around the peak of the hydrograph during several high flow events throughout the year. We
performed watershed-specific estimates of DOC flux using the "rloadest" package (Lorenz et al.,
2015) in R (version 3.2.5, R Core Team, 2016), which replicates functions developed in the U.S.
Geological Survey load-estimator program, LOADEST (Runkel et al., 2004). LOADEST is a
multiple-regression adjusted maximum likelihood estimation model that calibrates a regression
between measured constituent values and stream flow across seasons and time and then fits it to
combinations of coefficients representing nine predetermined models of constituent flux. To
account for potentially small sample size, the best model was selected using the second order
Akaike Information Criterion (AICc) (Akaike, 1981; Hurvich and Tsai, 1989). Input data were
log-transformed to avoid bias and centered to reduce multicollinearity. For additional details on
model selection, see Supplemental Table S3.1.
**2.6 Optical characterization of DOM**

Prior to May 2014, absorbance measures of water samples (n= 99) were conducted on a

Varian Cary-50 (Varian, Inc.) spectrophotometer at the BC Ministry of the Environment
Technical Services Laboratory (Victoria, BC, Canada) to determine specific UV absorption at
254 nm (SUVA$_{254}$). After May 2014, we conducted optical characterization of DOM by
absorbance and fluorescence spectroscopy at the Hakai Institute field station (Calvert Island, BC,
Canada) using an Aqualog fluorometer (Horiba Scientific, Edison, New Jersey, USA). Strongly
absorbing samples (absorbance units > 0.2 at 250 nm) were diluted prior to analysis to avoid
excessive inner filter effects (Lakowicz, 1999). Samples were run in 1 cm quartz cells and
scanned from 220-800 nm at 2 nm intervals to determine SUVA$_{254}$ as well as the spectral slope
ratio ($S_R$). SUVA$_{254}$ has been shown to positively correlate with increasing molecular aromaticity
associated with the fulvic acid fraction of DOM (Weishaar et al., 2003), and is calculated by
dividing the Decadic absorption coefficient at 254 nm by DOC concentration (mg C L$^{-1}$). To
account for potential Fe interference with absorbance values, we corrected SUVA$_{254}$ values by Fe
concentration according the method described in Poulin et al., (2014). $S_R$ has been shown to
negatively correlate with molecular weight (Helms et al., 2008), and is calculated as the ratio of
the spectral slope from 275 nm to 295 nm ($S_{275-295}$) to the spectral slope from 350 nm to 400 nm
($S_{350-400}$).

We measured excitation and emission spectra (as excitation emission matrices, EEMs) on

samples every three weeks from January to July 2016 (n= 63). Samples were run in 1 cm quartz
cells and scanned from excitation wavelengths of 230-550 nm at 5nm increments, and emission
wavelengths of 210-620 nm at 2 nm increments.  The Horiba Aqualog applied the appropriate
instrument corrections for excitation and emission, inner filter effects, and Raman signal
calibration. We calculated the Fluorescence Index and Freshness Index for each EEM.  The
Fluorescence Index is often used to indicate DOM source, where higher values are more
indicative of microbial-derived sources of DOM and lower values indicate more terrestrial-
derived sources (McKnight et al., 2001), and is calculated as the ratio of emission intensity at
450 nm to 500 nm, at an excitation of 370 nm.  The Freshness Index is used to indicate the
contribution of authochthonous or recently microbial-produced DOM, with higher values
suggesting greater autochthony (i.e., microbial inputs), and is calculated as the ratio of emission
intensity at 380 nm to the maximum emission intensity between 420 nm and 435 nm, at
excitation 310 nm (Wilson and Xenopoulos, 2009).

To further characterize features of DOM composition, we performed parallel factor

analysis (PARAFAC) using EEM data within the drEEM toolbox for Matlab (Mathworks, MA,
USA) (Murphy et al., 2013). PARAFAC is a statistical technique used to decompose the
complex mixture of the fluorescing DOM pool into quantifiable, individual components
(Stedmon et al., 2003). We detected a total of six unique components, and validated the model
using core consistency and split-half analysis (Murphy et al., 2013; Stedmon and Bro, 2008).
Components with similar spectra from previous studies were identified using the online
fluorescence repository, OpenFluor (Murphy et al., 2014), and additional components with
similar peaks were identified through literature review. Since the actual chemical structure of
fluorophores is unknown, we used the concentration of each fluorophore as maximum
fluorescence of excitation and emission in Raman Units ($F_{max}$) to derive the percent contribution
of each fluorophore component to total fluorescence. Relationships between PARAFAC
components were also evaluated using Pearson correlation coefficients in the R package "Hmisc"
(Harrell et al., 2016).
**2.7 Evaluating relationships in DOC concentration and DOM composition with stream**
**discharge and temperature**

We used linear mixed effects models to assess the relationship between DOC

concentration or DOM composition ($\delta^{13}$C-DOC, $S_R$, SUVA$_{254}$, Fluorescence Index, Freshness
Index, PARAFAC components), stream discharge, and stream temperature. Analysis was
performed in R using the nlme package (Pinheiro et al., 2016). Watershed was included as a
random intercept to account for repeat measures on each watershed. For some parameters, a
random slope of either discharge or temperature was also included based on data assessment and
model selection. Model selection was performed using AIC to compare models fit using
Maximum Likelihood (ML) (Burnham and Anderson, 2002; Symonds and Moussalli, 2010). The
final model was fit using Restricted Maximum Likelihood (REML). Marginal $R^2$, which
represents an approximation of the proportion of the variance explained by the fixed factors
alone, and conditional $R^2$, which represents an approximation of the proportion of the variance
explained by both the fixed and random factors, were calculated based on the methods described
in Nakagawa and Schielzeth (2013) and Johnson (2014).
**2.8 Redundancy analysis: Relationships between DOC concentration, DOM composition,**
**and watershed characteristics**
We evaluated relationships between stream water DOC and watershed characteristics by
relating DOC concentration and measures of DOM composition to catchment attributes using
redundancy analysis (RDA; type 2 scaling) in the package rdaTest (Legendre and Durand, 2014)
in R (version 3.2.2, R Core Team, 2015). To maximize the amount of information available, we
performed RDA analysis on samples collected from January to July 2016, and therefore included
all parameters of optical characterization (i.e., all PARAFAC components and spectral indices).
We assessed the collinearity of DOM compositional variables using a variance inflation factor
(VIF) criteria of > 10, which resulted in the removal of PARAFAC components C2, C3, and C5
prior to RDA analysis. Catchment attributes for each watershed included average slope, percent
area of lakes, percent area of wetlands, average depth of mineral soil, and average depth of
organic soil. Relationships between variables were linear, so no transformations were necessary
and variables were standardized prior to analysis. To account for repeat monthly measures per
watershed and potential temporal correlation associated with monthly sampling, we included
sample month as a covariable ("partial-RDA"). To test whether the RDA axes significantly
explained variation in the dataset, we compared permutations of residuals using ANOVA (9,999
iterations; test.axes function of rdaTest).
**3. Results**
**3.1 Hydrology**

We present work for water year 2015 (WY2015; October 1, 2014 – September 30, 2015)

and water year 2016 (WY2016; October 1, 2015 – September, 30, 2016). Annual precipitation
for both water years was lower than historical mean annual precipitation (WY2015= 2661 mm;
WY2016= 2587 mm). It is worth noting that mean annual precipitation at our rain gauge location
(2890 mm $yr^{-1}$, elevation = 16 m) is substantially lower than the average amount received at
higher elevations, which from 1981-2010 was approximately 5027 mm yr[-1] at an elevation of
1000m within our study area.  This area receives a very high amount of annual rainfall
(http://data.worldbank.org) but also experiences strong seasonal variation, with an extended wet
period from fall through spring, and a much shorter, typically drier period during summer. In
WY2015 and WY2016, 86-88% of the annual precipitation on Calvert Island occurred during the
8-months of wetter and cooler weather between September and April (~ 75% of the year),
designated the "wet period" (WY2015 wet= 2388 mm, average air temp= 7.97°C; WY2016 wet=
2235 mm; average air temp= 7.38°C). The remaining annual precipitation occurred during the
drier and warmer summer months of May – August, designated the "dry period" (WY2015 dry=
314 mm, average air temp= 13.4°C; WY2016 dry= 352 mm, average air temp= 13.1°C). Overall,
although WY2015 was slightly wetter than WY2016, the two years were comparable in relative
precipitation during the wet versus dry periods.
Stream discharge (Q) responded rapidly to rain events and as a result, closely tracked
patterns in total precipitation (Fig. 2). Total Q for all watersheds was on average 22% greater for
the wet period of WY2015 (total Q= 223.02 * $10^6$; range= 5.13 * $10^6$ – 111.51 * $10^6$ m$^3$)
compared to the wet period of WY2016 (total Q= 182.89 * $10^6$; range= 4.17 * $10^6$ – 91.45 * $10^6$
m$^3$). Stream discharge and stream temperature were significantly different for wet versus dry
periods (Mann-Whitney tests, p< 0.0001).
**3.2 Temporal and spatial patterns in DOC concentration, yield and flux**
Stream waters were high in DOC concentration relative to the global average for DOC
concentration in freshwater discharged directly to the ocean (average DOC for Calvert and
Hecate Islands = 10.4 mg L$^{-1}$, std= 3.8; average global DOC= ~ 6 mg L$^{-1}$) (Meybeck, 1982;
Harrison et al., 2005) (Table 1; Fig. 3). Q-weighted average DOC concentrations were higher
than average measured DOC concentrations (11.1 mg L$^{-1}$, Table 1), and also resulted in slightly
different ranking of the watersheds for highest to lowest DOC concentration. Within watersheds,
flow-weighted DOC concentrations ranged from a low of 8.4 mg L$^{-1}$ (watershed 693) to a high of
19.3 mg L$^{-1}$ (watershed 819), and concentrations were significantly different between watersheds
(Kruskal-Wallis test, p <0.0001). Seasonal variability tended to be higher in watersheds where
DOC concentration was also high (watersheds 626, 819, and 844) and lower in watersheds with
greater lake area (watersheds 1015 and 708) (Table 1; box plots, Figure 3). On an annual basis,
DOC concentrations generally decreased through the wet period, and increased through the dry
period, and concentrations were significantly lower during the wet period compared to the dry
period (Mann-Whitney test, p= 0.0123). Results of our linear mixed effects (LME) model
(Supplemental Table S6.1) indicate that DOC concentration was positively related to both
discharge ($b$= 0.613, p< 0.001) and temperature ($b$= 0.162, p= 0.011) (model conditional $R^2$=
0.57, marginal $R^2$= 0.09).

Annual and monthly DOC yields are presented in Table 1. For the total period of

available Q (October 1, 2014 - April 30, 2016; 19 months), areal (all watersheds) DOC yield was
52.3 Mg C km$^{-2}$ (95% CI= 45.7 to 68.2 Mg C km$^{-2}$) and individual watershed yields ranged from
24.1 to 43.6 Mg C km$^{-2}$. For WY2015, areal annual DOC yield was 33.3 Mg C km$^{-2}$ yr$^{-1}$ (95%
CI= 28.9 to 38.1 Mg C km$^{-2}$ yr$^{-1}$).  Total monthly rainfall was strongly correlated with monthly
DOC yield (Fig. 4), and average monthly yield for the wet period (3.35 Mg C km$^{-2}$ mo$^{-1}$; 95%
CI= 2.94 to 4.40 Mg C km$^{-2}$ mo$^{-1}$) was significantly greater than average monthly yield during
the dry period (0.50 Mg C km$^{-2}$ mo$^{-1}$; 95% CI= 0.41 to 0.62 Mg C km$^{-2}$ mo$^{-1}$) (Mann-Whitney
test, p< 0.0001).
Across our study watersheds, DOC flux generally increased with increasing watershed
area (Fig. 5). In WY2015, total DOC flux for all watersheds included in our study was 1562 Mg
C (95% CI= 1355 to 1787 Mg C), and individual watershed flux ranged from ranged from 82 to
276 Mg C. DOC flux was significantly different in wet versus dry periods (Mann-Whitney test, p
< 0.0001). Overall, 94% of the export in WY2015 occurred during the wet period, and export for
the wet period of WY2015 was lower than export for the wet period of WY2016 (Fig. 5).
**3.3 Temporal and spatial patterns in DOM composition**
The stable isotopic composition of dissolved organic carbon ($\delta^{13}$C-DOC) was relatively
tightly constrained over space and time (average $\delta^{13}$C-DOC= -26.53‰, std= 0.36; range= -
27.67‰ to -24.89‰). Values of $S_R$ were low compared to the range typically observed in surface
waters (average $S_R$= 0.78, std= 0.04; range= 0.71 to 0.89) and Fe-corrected SUVA$_{254}$ values
were at the high end of the range compared to most surface waters (average SUVA$_{254}$ for Calvert
and Hecate Islands= 4.42 L mg$^{-1}$ m$^{-1}$, std= 0.46; range of SUVA$_{254}$ in surface waters = 1.0 to 5.0
L mg$^{-1}$ m$^{-1}$) (Spencer et al., 2012). Values for both Fluorescence Index (average Fluorescence
Index= 1.36, std= 0.04; range= 1.30 to 1.44) and Freshness Index (average Freshness Index=
0.46, std= 0.02; range= 0.41 to 0.49) were relatively low compared to the typical range found in
surface waters (Fellman et al., 2010; Hansen et al., 2016). Differences between watersheds were
observed for $\delta^{13}$C-DOC (Kruskal-Wallis test, p= 0.0043), $S_R$ (Kruskal-Wallis test, p= 0.0001),
Fluorescence Index (Kruskal-Wallis test, p= 0.0030), and Freshness Index (Kruskal-Wallis test,
p= 0.0099), but watersheds did not differ in SUVA$_{254}$ (Kruskal-Wallis test, p= 0.4837).
We did not observe an obvious seasonal trend in $\delta^{13}$C-DOC (Fig. 3), but LME model
results (Supplemental Table S6.1) indicate that $\delta^{13}$C-DOC declined with increasing discharge
($b$= -0.049, p= 0.014) and stream temperature ($b$= -0.024, p< 0.001) (model conditional R$^2$=
0.35, marginal $R^2 = 0.10$). In contrast, although $SUVA_{254}$ appeared to exhibit a general seasonal
trend of values increasing over the wet period and decreasing over the dry period, $SUVA_{254}$ was
not significantly related to either discharge or stream temperature in the LME model results. $S_R$
also appeared to fluctuate seasonally, with lower values during the wet season and higher values
during the dry season. $S_R$ was negatively related to discharge ($b = -0.026$, $p < 0.001$) and
positively related to the interaction between discharge and stream temperature ($b = 0.0015$, $p <$
$0.001$) (model conditional $R^2 = 0.62$, marginal $R^2 = 0.28$). Freshness Index was negatively related
to stream temperature ($b = -0.003$, $p = 0.008$) (model conditional $R^2 = 0.59$, marginal $R^2 = 0.23$),
while Fluorescence Index was not significantly related to either discharge or stream temperature.
**3.4 PARAFAC characterization of DOM**

Six fluorescence components were identified through PARAFAC ("C1" through "C6")

(Table 2). Additional details on PARAFAC model results are provided in Supplemental Table
S4.1, Fig. S4.2, and Fig. S4.3. Of the six components, four were found to have close spectral
matches in the OpenFluor database (C1, C3, C5, C6; minimum similarity score > 0.95), while
the remaining two (C2 and C4) were found to have similar peaks represented in the literature.
The first four components (C1 through C4) are described as terrestrial-derived, whereas
components C5 and C6 are described as autochthonous or microbially-derived (Table 2). In
general, the rank order of each components' percent contribution to total fluorescence was
maintained over time, with C1 comprising the majority of total fluorescence across all
watersheds (Fig. 6).

Across watersheds, components fluctuated synchronously over time and variation

between watersheds was relatively low, although slightly more variation between watersheds
was observed during the beginning of the dry period relative to other times of the year (Fig 6).
The percent contributions of components C1, C3, C5 and C6 to total fluorescence were not
significantly different across watersheds (for all components Kruskal-Wallis test, $p > 0.05$),
however percent composition of both C2 and C4 were different (Kruskal-Wallis test, $p = 0.0306$
and $p= 0.0307$, respectively) and higher for watersheds 819 and 844 relative to the other
watersheds (Supplemental Fig. S4.4).

PARAFAC components exhibited significant relationships with stream discharge and

stream temperature, although predicted changes (beta, or $b$) in fluorescence components with
discharge and/or stream temperature were small (Supplemental Table S6.2). C3 increased with
discharge ($b= 0.006$, $p= 0.003$), whereas C2, C4, and C5 decreased with discharge (C2: $b= -$
$0.005$, $p= 0.022$; C4: $b= -0.008$, $p= 0.002$; C5: $b= -0.008$, $p= 0.002$). C1, C4, and C6 increased
with temperature (C1: $b= 0.001$, $p= 0.050$; C4: $b= 0.003$, $p< 0.001$; C6: $b= 0.005$, $p= 0.005$),
while both C3 and C5 decreased with temperature (C3: $b= -0.003$, $p= 0.003$; C5: $b= -0.003$, $p=$
$0.027$). Conditional $R^2$ values for the models ranged from 0.28 to 0.69, while marginal $R^2$ ranged
from 0.20 to 0.46. Overall, greater changes in component contribution to total fluorescence were
observed with changes in discharge relative to changes in stream temperature.
**3.5 Relationships between watershed characteristics, DOC concentrations, and DOM**
**composition**

Results of the partial-RDA (type 2 scaling) were significant in explaining variability in

DOM concentration and composition (semi-partial $R^2= 0.33$, $F= 7.90$, $p< 0.0001$) (Fig. 7). Axes
1 through 3 were statistically significant at $p< 0.001$, and the relative contribution of each axis to
the total explained variance was 47%, 30%, and 22%, respectively. Additional details on the
RDA test are provided in Supplemental Figs. S5.1-S5.2 and Tables S5.3 – S5.5. Axis 1 described
a gradient of watershed coverage by water-inundated ecosystem types, ranging from more
wetland coverage to more lake coverage. Total lake coverage (area) and mean mineral soil
material thickness showed a strong positive contribution, and wetland coverage (area) showed a
strong negative contribution to this axis. Freshness Index, Fluorescence Index, $S_R$ and
fluorescence component C6 were positively correlated with Axis 1, while component C4 showed
a clear negative correlation. Axis 2 described a subtler gradient of soil material thickness ranging
from greater mean organic soil material thickness to greater mean mineral soil material
thickness. DOC concentration, $\delta^{13}$C-DOC, SUVA$_{254}$, and fluorescence component C1 all showed
a strong, positive correlation with Axis 2. Axis 3 described a gradient of watershed steepness,
from lower gradient slopes with more wetland area and thicker organic soil material to steeper
slopes with less developed organic horizons. Average slope contributed negatively to Axis 3 (see
Supplemental Table S5.5), followed by positive contributions from both wetland area and
thickness of organic soil material. $\delta^{13}$C-DOC showed the most positive correlation with Axis 3,
whereas fluorescence components C1 and C4 showed the most negative.
**4. Discussion**
**4.1 DOC export from small catchments to the coastal ocean**

In comparison to previous studies, our estimate of freshwater DOC yields from Calvert

and Hecate Island watersheds are in the upper range predicted for this region based on global
models (Mayorga et al., 2010) and DOC exports quantified for southeastern Alaska (D'Amore et
al., 2015a; D'Amore et al., 2016; Stackpoole et al., 2017). Compared to watersheds of similar
size, DOC yields from Calvert and Hecate Island watersheds are some of the highest observed
(see reviews in Hope et al., 1994; Alvarez-Cobelas et al., 2012), including DOC yields
determined from many tropical rivers, despite the fact that tropical rivers have been shown to
export very high DOC (e.g., Autuna River, Venezuela, DOC yield= 56,946 kg C km$^{-2}$ yr$^{-1}$;
Castillo et al., 2004), and are often regarded as having disproportionately high carbon export
compared to temperate and Arctic rivers (Aitkenhead and McDowell, 2000; Borges et al., 2015).
Our estimates of DOC yield are comparable to, or higher than, previous estimates from high-
latitude catchments of similar size that receive high amounts of precipitation and contain
extensive organic soils and wetlands (e.g. Naiman, 1982 (DOC yield= 48,380 kg C km$^{-2}$ yr$^{-1}$);
Brooks et al., 1999 (DOC yield= 20,300 kg C km$^{-2}$ yr$^{-1}$); Ågren et al., 2007 (DOC yield= 32,043
kg C km$^{-2}$ yr$^{-1}$)). However, many of these catchments represent low (first or second) order
headwater streams that drain to higher order stream reaches, rather than directly to the ocean.
Although headwater streams have been shown to export up to 90 % of the total annual carbon in
stream systems (Leach et al., 2016), significant processing and loss typically occurs during
downstream transit (Battin et al., 2008).

Over much of the incised outer coast of the CTR, small rainfall-dominated catchments

contribute high amounts of freshwater runoff to the coastal ocean (Royer, 1982; Morrison et al.,
2012; Carmack et al., 2015). Small mountainous watersheds that discharge directly to the ocean
can exhibit disproportionately high fluxes of carbon relative to watershed size, and in aggregate
may deliver more than 50% of total carbon flux from terrestrial systems to the ocean (Milliman
and Syvitski, 1992; Masiello and Druffel, 2001). Extrapolating our estimate of annual DOC yield
from Calvert and Hecate Island watersheds to the entire hypermaritime subregion of British
Columbia's CTR (29,935 km$^2$), generates an estimated annual DOC flux of 0.997 Tg C yr$^{-1}$
(0.721 to 1.305 Tg C yr$^{-1}$ for our lowest to highest yielding watersheds, respectively), with the
caveat that this estimate is rudimentary and does not account for spatial heterogeneity in
controlling factors such as wetland extent, topography, watershed size. Regional comparisons
estimate that Southeast Alaska (104,000 km$^2$), at the northern range of the CTR, exports
approximately 1.25 Tg C yr$^{-1}$ (Stackpoole et al., 2016), while south of the perhumid CTR, the
wet northwestern United States and its associated coastal temperate rainforests export less than
0.153 Tg C yr$^{-1}$ as DOC (reported as TOC, Butman et al., 2016). This suggests that the
hypermaritime coast of British Columbia plays an important role in the export of DOC from
coastal temperate rainforest ecosystems of western North America, in a region that is already
expected to contribute high quantities of DOC to the coastal ocean.
**4.2. Seasonal variability in DOC export**

Despite having an ocean-moderated climate compared to continental interiors, the study

area experiences seasonal patterns in precipitation dominated by a longer wet period and a
shorter, drier period. Flashy stream hydrographs indicate that hydrologic response times for
Calvert and Hecate Island watersheds are rapid, presumably as a result of small catchment size,
high drainage density, and relatively shallow soils with high hydraulic conductivity (Gibson et
al., 2000; Fitzgerald et al., 2003). Rapid runoff is presumably accompanied by rapid increases in
water tables and lateral movement of water through shallow soil layers rich in organic matter
(Fellman et al., 2009b; D'Amore et al., 2015b). During drier periods DOC pools increase in soils
and are flushed to streams when water tables rise (Boyer et al., 1996).

Precipitation is a well-established driver of stream DOC export (Alvarez-Cobelas et al.,

2012), particularly in systems containing organic soils and wetlands (Olefeldt et al., 2013; Wallin
et al., 2015; Leach et al., 2016). Frequent, high intensity precipitation events and short residence
times are expected to result in pulsed exports of stream DOC that are rapidly shunted
downstream, thus reducing time for in-stream processing (Raymond et al., 2016). On Calvert and
Hecate Islands, the combination of high rainfall, rapid runoff, and abundant sources of DOC
from organic-rich soils, wetlands, and forests, result in high DOC fluxes. The process of "DOC
flushing" has been shown to increase stream water DOC during higher flows in coastal and
temperate watersheds (e.g., Sanderman et al., 2009; Deirmendjian et al., 2017). In our study, the
relationship between DOC concentration and discharge varied by watershed (see Supplemental
Fig. S6.1), but overall DOC concentrations increased with both discharge and temperature. This
indicates that while watershed characteristics are important for influencing the magnitude and
variability of DOC concentrations and export, the hydrologic coupling of precipitation and
discharge with seasonal production and availability of DOC is an overarching driver of DOC
export (Fasching et al., 2016).
**4.3 DOM character: Sources and variability**
Measures of DOM composition from Calvert and Hecate Islands suggest that carbon and
organic matter exported from these systems is highly terrestrial. Values for $\delta^{13}$C-DOC were
relatively constrained, suggesting terrestrial carbon sources from C3 plants and soils were the
dominant input to catchment stream water DOM (Finlay and Kendall, 2007). Measures of $S_R$ and
SUVA$_{254}$ were typical of environments that export large quantities of high molecular weight,
highly aromatic DOM such as some tropical rivers (e.g., Lambert et al., 2016; Mann et al., 2014),
streams draining wetlands (e.g., Ågren et al., 2008, Austnes et al., 2010), or streams draining
small undisturbed catchments comprised of mixed forest and wetlands (e.g. Wickland et al.,
2007; Fellman et al., 2009a; Spencer et al., 2010, Yamashita et al., 2011). This suggests the
majority of the DOM pool is comprised of larger molecules that have not been extensively
chemically or biologically degraded through processes such as microbial utilization or
photodegradation, and therefore are potentially more biologically available (Amon and Benner,

1996).

Seasonal variability in DOM composition may be attributed to seasonal changes in
biological activity and shifting flow paths that affect hydrologic interactions with different DOM
source materials (Fellman et al., 2009b). On Calvert and Hecate Island watersheds, some
measures of DOM composition, such as $\delta^{13}$C-DOC and $S_R$, exhibited seasonal patterns. In our
study, discharge was significantly related to $\delta^{13}$C-DOC and $S_R$, with higher discharge resulting in
more terrestrial-like DOM (i.e., more depleted $\delta^{13}$C-DOC and lower $S_R$) as saturated conditions
promote the mobilization of a wider range of DOM from soil material (McKnight et al., 2001;
Kalbitz et al., 2002). This is similar to findings of Sanderman et al. (2009), who observed distinct
relationships between discharge and both $\delta^{13}$C-DOC and SUVA$_{254}$, and postulated that during
the rainy season, hillslope flushing shifts DOM sources to more aged soil organic material
because plant productivity is not rapid enough to meet microbial demand, forcing microbes to
switch to metabolizing more aged DOM within soils. It has also been shown that rising water
tables can establish strong hydraulic gradients that initiate and sustain prolonged increases in
metrics like SUVA$_{254}$, until the progressive drawdown of upland water tables constrain flow
paths (Lambert et al., 2013).
During the drier and warmer period, DOM decreased in molecular weight ($S_R$) and
Freshness Index, as well as increased in C6, a component comprised of protein-like composition.
This suggests a shift in the source of DOM and/or increased contributions from less aromatic,
lower molecular weight material, such as DOM derived from increased terrestrial primary
production (Berggren et al., 2010), and perhaps deeper flow paths that contribute to mineral
binding and export of older, more processed terrestrial material (McKnight et al., 2001; van Hees
et al., 2005). Proportions of fluorescence components were generally consistent across
watersheds during the dry period, but diverged during the wet period, further suggesting that
water table draw down and unsaturated soils lead to more diverse flow paths and hydrologic
interaction with different sources of DOM.

Interestingly, more depleted values of $\delta^{13}$C-DOC were also related to warmer

temperature. The positive relationship between $\delta^{13}$C-DOC and both discharge and temperature,
as well as the overall low variability in $\delta^{13}$C-DOC, suggests that the availability or production of
terrestrial DOM is enough to keep up with microbial demand, allowing the supply of terrestrial
material to remain relatively seasonally consistent. The positive relationship of temperature and
Freshness Index, as well as with C1 and C4, further suggests that warmer periods contribute to a
fresh supply of terrestrial material available for microbial degradation and export (Fellman et al.,
2009a; Fasching et al., 2016).

The interaction of sources and flow paths during wet versus dry periods may have

important consequences for the downstream fate of this material. For example, biological
utilization of DOM is influenced by its composition (e.g. Judd et al., 2006; Fasching et al.,
2014), therefore differences in the nature of DOM exports will likely alter the downstream fate
and ecological role of freshwater-exported DOM. The majority of the fluorescent DOM pool was
comprised of C1, which is described as humic-like, less-processed terrestrial soil and plant
material (see Table 2). This may represent a relatively fresh, seasonally-consistent contribution
of terrestrial subsidy from streams to the coastal ecosystem, which is relatively lower in carbon
and nutrients throughout much of the year (Whitney et al., 2005; Johannessen et al., 2008). For
example, pulsed contributions of less-processed humic material exported from rivers to lakes
have been shown to stimulate bacterial production (Bergström and Jansson, 2000). While
previous studies have suggested that bacteria prefer autochthonous carbon sources, they readily
utilize allochthonous terrestrial DOC subsidies (Bergström and Jansson, 2000; Kritzberg et al.,
2004; McCallister and Giorgio, 2008; Berggren et al., 2010), enabling humic and fulvic material
to fuel a low but continuous level of bacterial productivity after more labile sources have been
consumed (Guillamette and Giorgio, 2011). In addition, although the tryptophan-like component
C6, represents a minor, more variable proportion of total fluorescence in comparison to the more
humic compounds such as C1, even a small proteinaceous fraction of the overall DOM pool can
play a major role in overall bioavailability and bacterial utilization of DOM (Berggren et al.,
2010; Guillamette and Giorgio, 2011).
**4.4 Relationships between watershed attributes and exported DOM**

Previous studies have implicated wetlands as a major driver of DOM composition (e.g.,

Xenopoulos et al., 2003; Ågren et al., 2008; Creed et al., 2008), however the analysis of
relationships between Calvert and Hecate Island landscape attributes and variation in DOM
composition suggests that controls on DOM composition are more nuanced than beign solely
driven by the presence of wetlands. Ågren et al. (2008) found that when wetland area comprised
>10% of total catchment area, wetland DOM was the most significant driver of stream DOM
composition during periods of high hydrologic connectivity. Although wetlands comprise an
average of 37% of our study area, they do not appear to be the single leading driver of variability
in DOC concentration and DOM composition. Other factors, such as watershed slope, the depth
of organic and mineral soil materials, and the presence of lakes also appear to be influence DOC
concentration and DOM composition.

In these watersheds, soils with pronounced accumulations of organic matter are not

restricted to wetland ecosystems. Peat accumulation in wetland ecosystems results in the
formation of organic soils (Hemists), where mobile fractions of DOM accumulate under
saturated soil conditions and limited drainage, resulting in the enrichment of poorly
biodegradable, more stable humic acids (Stevenson, 1994; Marschner and Kalbitz, 2003).
Although Hemist soils comprise 27.8% of our study area, Folic Histosols, which form under
more freely drained conditions, such as steeper slopes, occur over an additional 25.7% of the
region (Supplemental S1.2). In freely drained organic soils, high rates of respiration can result in
further enrichment of aromatic and more complex molecules, and this material may be rapidly
mobilized and exported to streams (Glatzel et al., 2003). This suggests the importance of widely
distributed, alternative soil DOM source-pools, such as Folic Histosols and associated Podzols
with thick forest floors on hillslopes, available to contribute high amounts of terrestrial carbon
for export.

Although lakes make up a relatively small proportion of the total landscape area, their

influence on DOM export appears to be important. The proportion of lake area can be a good
predictor of organic carbon loss from a catchment since lakes often increase hydrologic
residence times and thus increase opportunities for biogeochemical processing (Algesten et al.,
2004; Tranvik et al., 2009). In our study, watersheds with a larger percentage of lake area
exhibited slower response following rain events (Supplemental Fig. S2.2), lower DOC yields,
and lake area was correlated with parameters that represent greater autochthonous DOM
production or microbial processing such as higher Freshness Index, $S_R$, Fluorescence Index, and
higher proportions of component C6. In contrast, watersheds with a high percentage of wetlands
contributed DOM that was more allocthonous in composition. Lakes are known to be important
landscape predictors of DOC, as increased residence time enables removal via respiration, thus
reducing downstream exports from lake outlets (Larson et al., 2007). The proximity of wetlands
and lakes within the catchment and their proximity to the watershed outlet can also play an
important role in the composition of DOM exports (Martin et al., 2006).

**5. Conclusions**


Previous work has demonstrated freshwater discharge is substantial along the coastal

margin of the North Pacific temperate rainforest, and plays an important role in processes such as
ocean circulation (Royer, 1982; Eaton and Moore, 2010). Our finding that small catchments in
this region contribute high yields of terrestrial DOC to coastal waters suggests that freshwater
inputs may also influence ocean biogeochemistry and food web processes through terrestrial
organic matter subsidies. Our findings also suggest that this region may be currently
underrepresented in terms of its role in global carbon cycling. Currently, there is no region-wide
carbon flux model for the Pacific coastal temperate rainforest or the greater Gulf of Alaska,
which would quantify the importance of this region within the global carbon budget. Our
estimates represent the hypermaritime outer-coast zone of the CTR, where subdued terrain, high
rainfall, ocean moderated temperatures and poor bedrock have generated a distinctive 'bog-
forest' landscape mosaic within the greater temperate rainforest (Banner et al. 2005). Even
within our geographically limited study area, we observed a range of DOC yields across
watersheds. To quantify regional scale fluxes of rainforest carbon to the coastal ocean, further
research will be needed to estimate DOC yields across complex spatial gradients of topography,
climate, hydrology, soils and vegetation. Long term changes in DOC flux have been observed in
many places (e.g., Worrall et al., 2004; Borken et al., 2011; Lepistö et al., 2014; Tank et al.,
2016) and continued monitoring of this system will allow us to better understand the underlying
drivers of export and evaluate future patterns in DOC yields. Coupled with current studies
investigating the fate of terrestrial material in ocean food webs, this work will improve our
understanding of coastal carbon patterns, and increase capacity for predictions regarding the
ecological impacts of climate change.

**Author Contributions**

The authors declare that they have no conflict of interest.

A.A. Oliver prepared the manuscript with contributions from all authors, designed analysis protocols, analyzed samples, performed the modeling and analysis for dissolved organic carbon fluxes, parallel factor analysis of dissolved organic matter composition, and all remaining statistical analyses. S.E. Tank assisted with designing the study and overseeing laboratory analyses, crafting the scope of the paper, and determining the analytical approach.

I. Giesbrecht led the initial DOC sampling design, helped coordinate the research team, oversaw routine sampling and data management, and led the watershed characterization.

M.C. Korver developed the rating curves, and conducted the statistical analysis of discharge measurement uncertainties and rating curve uncertainties. W.C. Floyd lead the hydrology component of this project, selected site locations, installed and designed the hydrometric stations, and developed the rating curves and final discharge calculations. C. Bulmer and P. Sanborn collected and analyzed soil field data and prepared the digital soils map of the watersheds. K.P. Lertzman conceived of and co-led the overall study of which this paper is a component, helped assemble and guide the team of researchers who carried out this work, provided input to each stage of the study.

**Acknowledgements**

This work was funded by the Tula Foundation and the Hakai Institute. The authors would like to thank many individuals for their support, including Skye McEwan, Bryn Fedje, Lawren McNab, Nelson Roberts, Adam Turner, Emma Myers, David Norwell, and Chris Coxson for sample collection and data management, Clive Dawson and North Road Analytical for sample

processing and data management, Keith Holmes for creating our maps, Matt Foster for database
development and support, Shawn Hateley for sensor network maintenance, Jason Jackson, Colby
Owen, James McPhail, and the entire staff at Hakai Energy Solutions for installing and
maintaining the sensors and telemetry network, and Stewart Butler and Will McInnes for field
support. Thanks to Santiago Gonzalez Arriola for generating the watershed summaries and
associated data products, and Ray Brunsting for overseeing the design and implementation of the
sensor network and the data management system at Hakai. Additional thanks to Lori Johnson
and Amelia Galuska for soil mapping field assistance, and Francois Guillamette for PARAFAC
consultation. Thanks to Dave D'Amore for inspiring the Hakai project to investigate aquatic
fluxes at the coastal margin and for technical guidance. Lastly, thanks to Eric Peterson and
Christina Munck who provided significant guidance throughout the process of designing and
implementing this study.

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

**Figure 1.** The location of Calvert Island, British Columbia, Canada, within the perhumid region
of the coastal temperate rainforest (right) and the study area on Calvert and Hecate Islands,
including the seven study watersheds, corresponding stream outlet sampling stations, and
location of the rain gauge (left). Characteristics of individual watersheds are described in Table
1212 1.

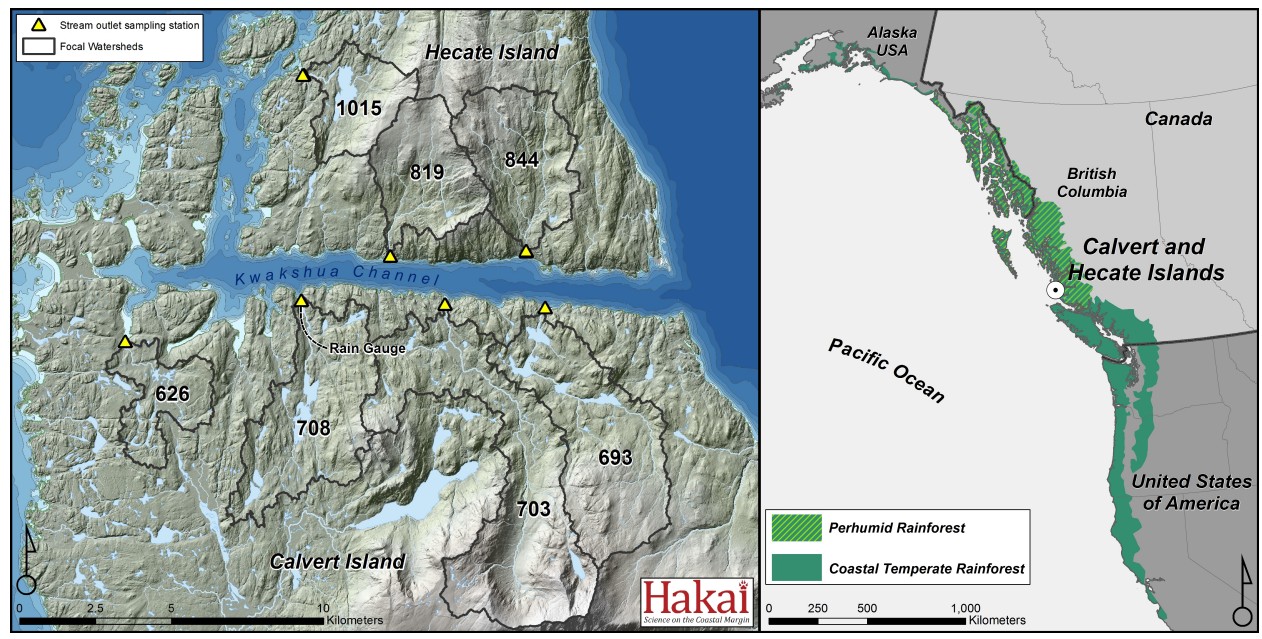


**Figure 2.** Hydrological patterns typical of watersheds located in the study area (a) the
hydrograph and precipitation record from Watershed 708 for the study period of October 1,
2015-April 30, 2016. Grey shading indicates the wet period (September 1-April 30) and the
unshaded region indicates the dry period (May 1-August 30) (b) Correlation of daily (24 hour)
areal runoff (discharge of all watersheds combined) to 48 hour total rainfall recorded at
watershed 708. For the period of study, comparisons of daily runoff to 48-hr rainfall
(runoff:rainfall mean= 0.92, std ±0.27) indicated rapid discharge response to rainfall.

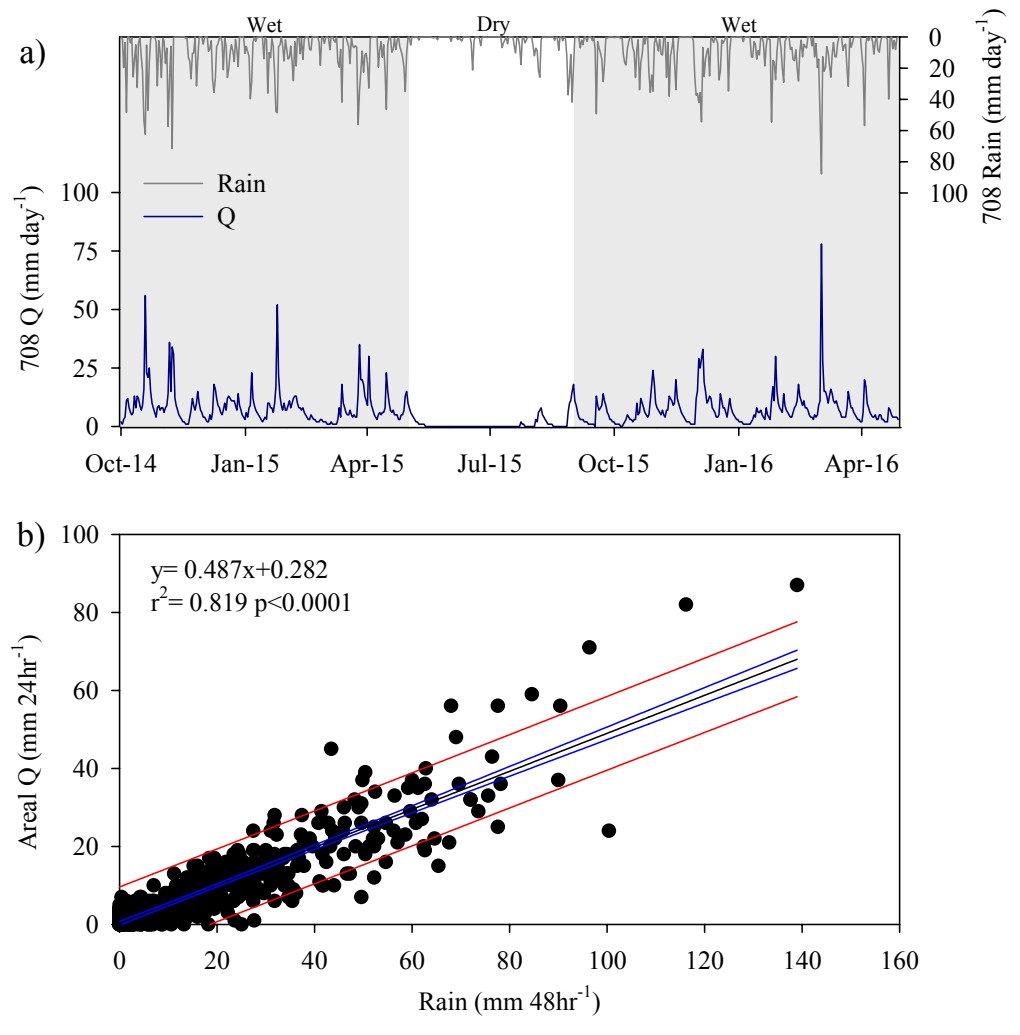


**Figure 3.** Seasonal (timelines, by date) and spatial (boxplots, by watershed) patterns in DOC
concentration and DOM composition for stream water collected at the outlets of the seven study
watersheds on Calvert and Hecate Islands. Boxes represent the 25th and 75th percentile, while
whiskers represent the 5th and 95th percentile. Daily precipitation and annual temperature are
shown in the top left panel.  Grey shading indicates the wet period (September 1-April 30) and
the unshaded region indicates the dry period of each water year.

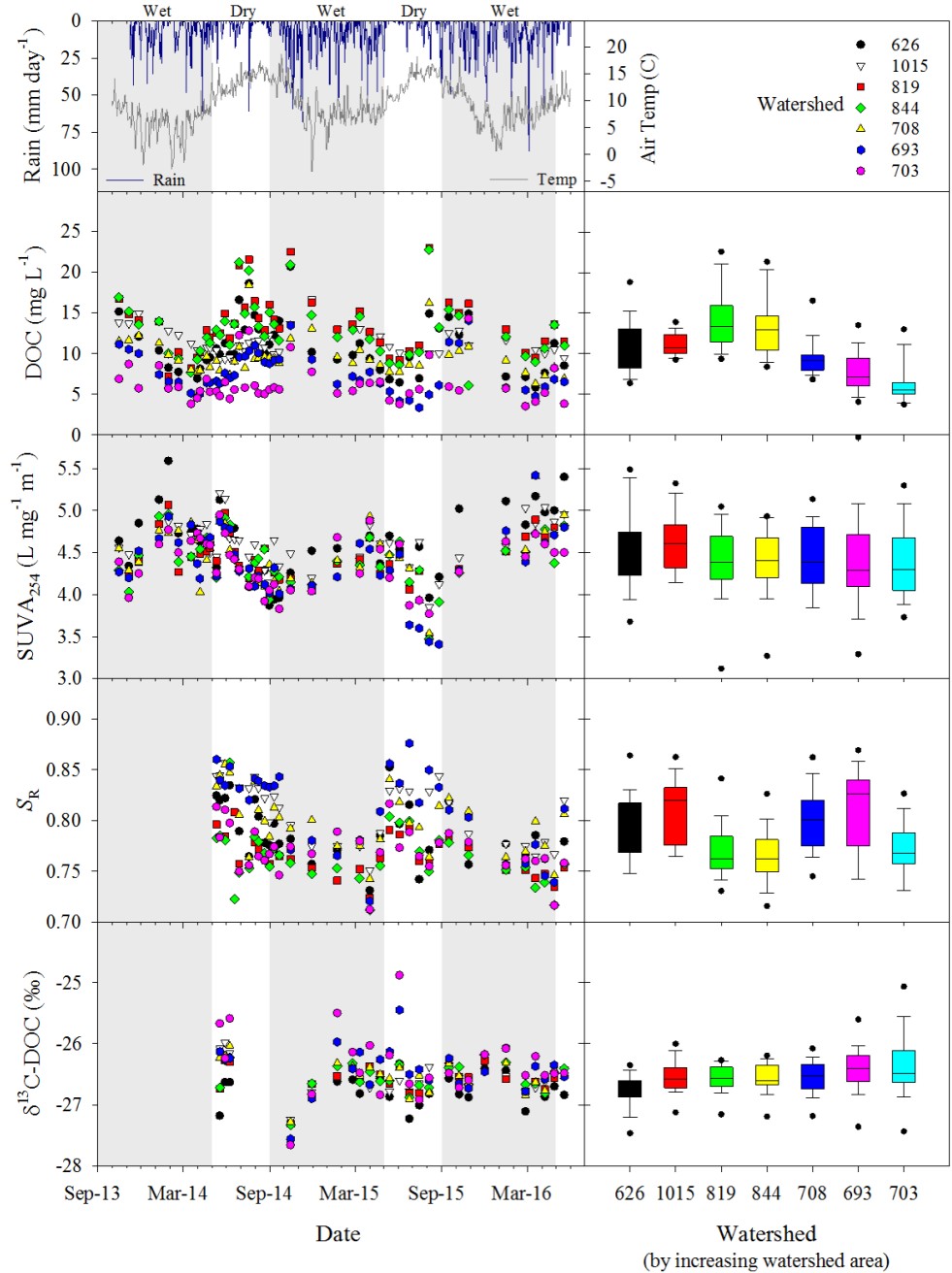


**Figure 4.** Monthly areal DOC yields and precipitation for water year 2015 (WY2015) and the
wet period (October 1-April 30) of water year 2016 (WY2016).  Error bars represent standard
error. Total rain and DOC yield were significantly correlated ($r^2$ = 0.77) and months of higher
rain produced higher DOC yields. In WY2015, the majority of DOC export (~94% of annual
flux) occurred during the wet period (~88% of annual precipitation).

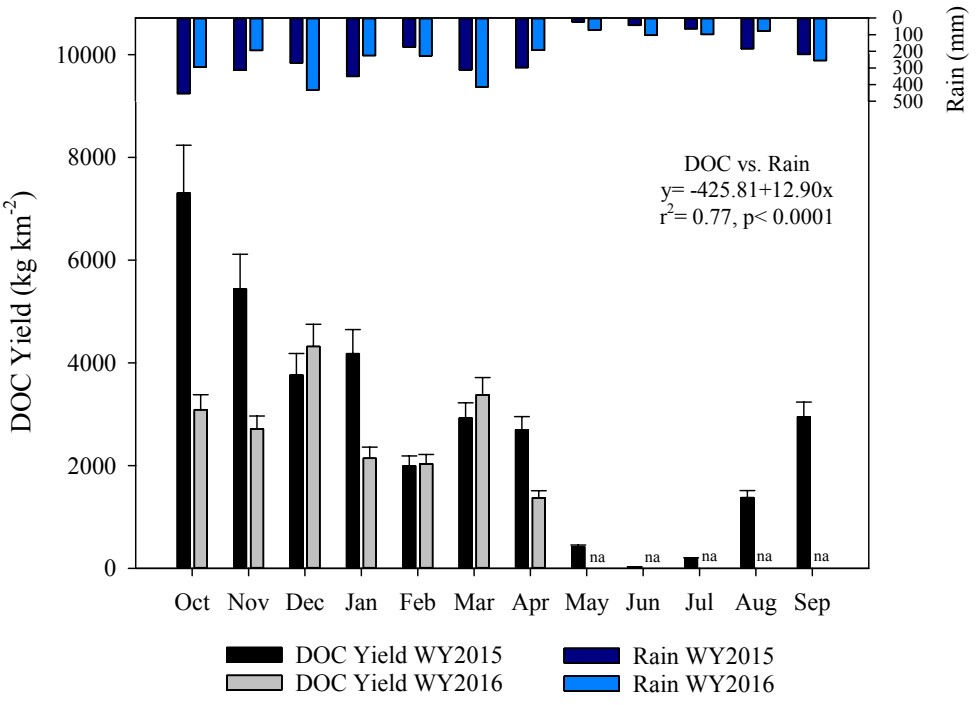


**Figure 5:** DOC fluxes and yields for the seven study watersheds and the total area of study
("areal", all watersheds combined) on Calvert and Hecate Islands for water year 2015 (WY2015;
Oct 1 - Sep 30), and October 1- April 30 of the wet period for water year 2015 (WY2015 wet)
and water year 2016 (WY2016 wet). Because DOC yields were only available for September in
WY2015, this month was excluded from the wet period totals in order to make similar
comparisons between years. Error bars represent standard error.

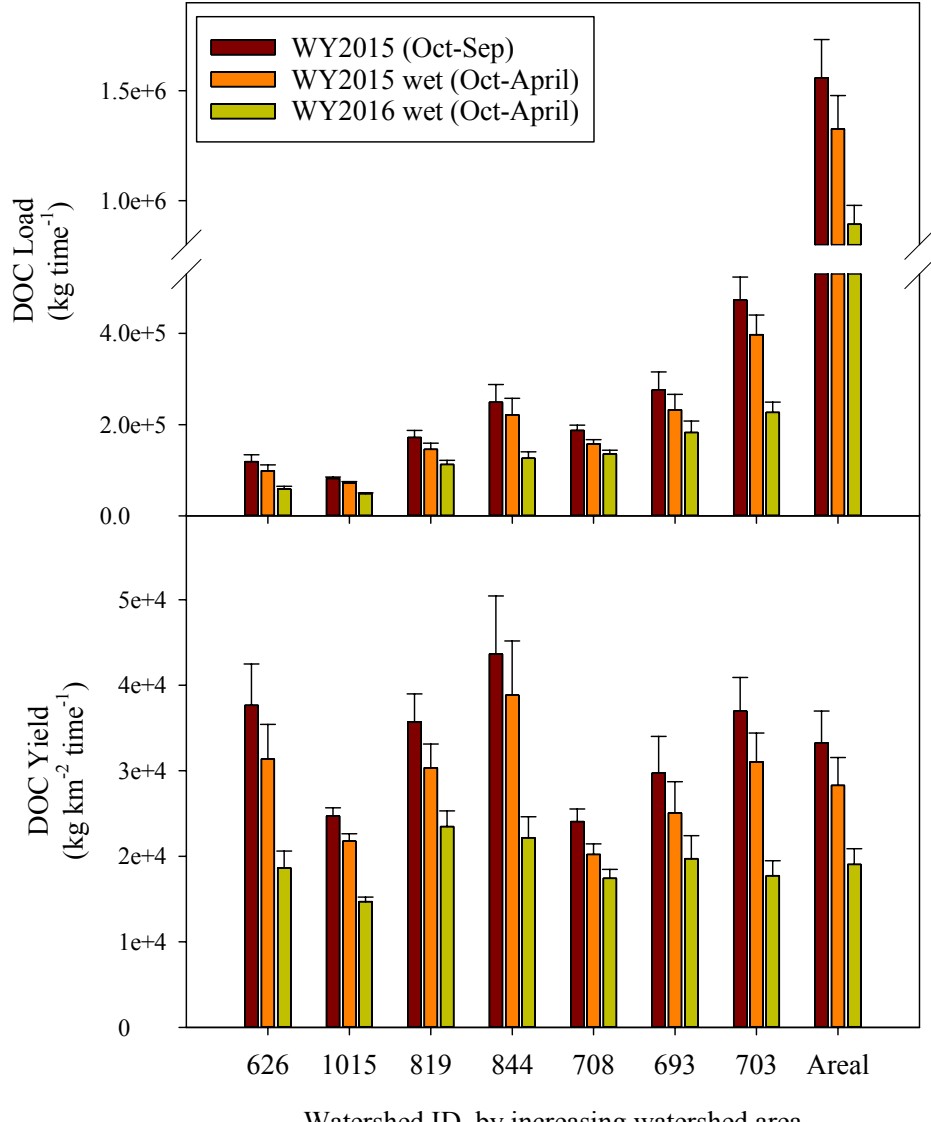

**Figure 6:** Percent contribution of the six components identified in parallel factor analysis
(PARAFAC) for samples collected every three weeks from January-July, 2016 from the seven
study watersheds on Calvert and Hecate Islands. The grey shading indicates the wet period and
**Figure 7:** Results from the partial-Redundancy analysis (RDA; type 2 scaling) of DOC
concentration and DOM composition versus watershed characteristics. Angles between vectors
represent correlation, i.e., smaller angles indicate higher correlation. Symbols represent different

watersheds, and numbers on symbols represent the sample month in 2016: 1= January, 2=
February, 3= March, 4= early April, 5= late April, and 6= May.

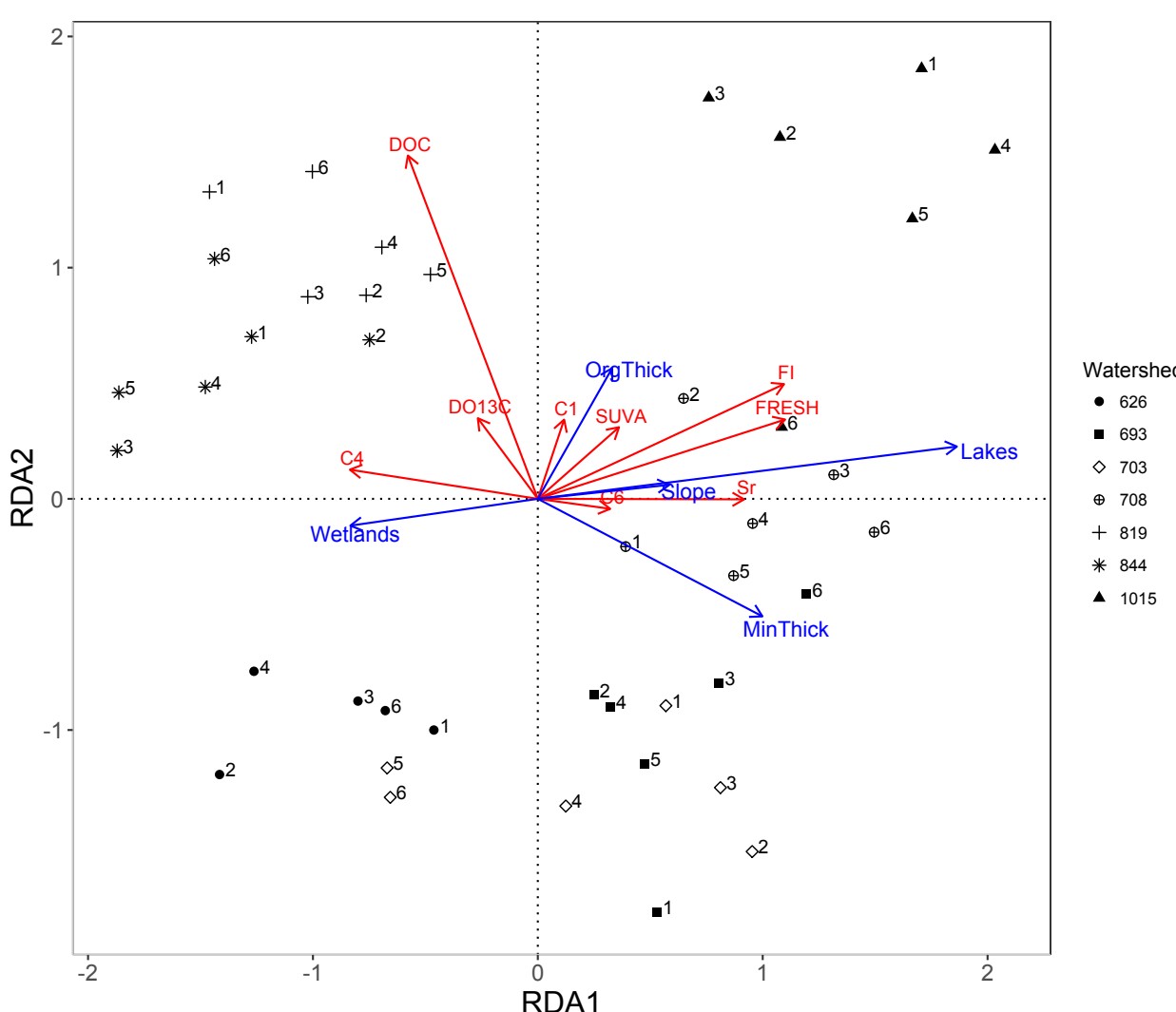


**Table 1:** Watershed characteristics, discharge, DOC concentrations, and DOC yields for the seven study watersheds on Calvert and
Hecate Islands. Additional details on the methods used to determine watershed characteristics can be found in Supplemental Material.

| Water-shed | Area (km$^2$) | Avg. Slope (%) | Lakes (% Area) | Wetlands (% Area) | Avg. Depth Organic Soils (cm) | Avg. Depth Mineral Soils (cm) | Total Q Yield* (mm) | DOC*[a] (mg L$^{-1}$) | Q-weighted Avg. DOC* (mg L$^{-1}$) | DOC Annual Yield[b] WY2015* (Mg C km$^{-2}$) | DOC Monthly Yield[b] Wet Season** (Mg C km$^{-2}$) | DOC Monthly Yield[b] Dry Season*** (Mg C km$^{-2}$) |
|---|---|---|---|---|---|---|---|---|---|---|---|---|
| 626 | 3.2 | 21.7 | 4.7 | 48.0 | 39.4 ±24.3 | 30.8 ±8.3 | 3673 | 11.0 ±3.5 | 15.3 | 37.7 (31.9 – 44.2) | 3.59 (3.05 – 4.18) | 0.62 (0.49 – 0.77) |
| 1015 | 3.3 | 34.2 | 9.1 | 23.8 | 39.5 ±17.2 | 33.7 ±8.6 | 3052 | 11.2 ±1.6 | 12.9 | 24.7 (23.6 – 25.8) | 2.56 (2.45 – 2.78) | 0.27 (0.25 – 0.28) |
| 819 | 4.8 | 30.1 | 0.3 | 50.2 | 37.9 ±19.1 | 29.8 ±5.7 | 3066 | 14.0 ±3.5 | 19.3 | 35.7 (31.7 – 40.2) | 3.80 (3.37 – 5.10) | 0.57 (0.48 – 0.67) |
| 844 | 5.7 | 32.5 | 0.3 | 35.2 | 35.4 ±18.0 | 29.1 ±6.4 | 4129 | 13.1 ±3.6 | 15.9 | 43.6 (34.2 – 54.9) | 4.24 (3.36 – 5.30) | 0.54 (0.36 – 0.77) |
| 708 | 7.8 | 28.5 | 7.5 | 46.3 | 36.2 ±19.7 | 29.9 ±6.0 | 3805 | 9.5 ±2.4 | 10.9 | 24.1 (22.2 – 26.0) | 2.67 (2.46 – 4.07) | 0.38 (0.34 – 0.43) |
| 693 | 9.3 | 30.2 | 4.4 | 42.8 | 35.4 ±16.1 | 30.2 ±6.4 | 5866 | 7.7 ±2.5 | 8.4 | 29.7 (25.9 – 34.0) | 3.19 (2.79 – 4.94) | 0.41 (0.32 – 0.52) |
| 703 | 12.8 | 40.3 | 1.9 | 24.3 | 37.3 ±16.5 | 35.8 ±13.4 | 6058 | 6.3 ±2.6 | 9.0 | 37.0 (32.5 – 42.0) | 3.48 (3.07 – 4.02) | 0.64 (0.52 – 0.77) |
| All | 46.9 | 32.7 | 3.7 | 37.1 | 37.4 ±17.7 | 32.2 ±9.2 | 4730 | 10.4 ±3.8 | 11.1 | 33.3 (28.9 – 38.1) | 3.35 (2.94 – 4.40) | 0.50 (0.41 – 0.62) |

\* Calculated for water year 2015 (WY2015; Oct 1, 2014-Sep 30, 2015)
\*\* Wet period average monthly yield calculated from October-April and September, WY2015 and October-April, WY2016
\*\*\* Dry period average monthly yield calculated from May-August, WY2015
[a] Mean ± standard deviation
[b] Total ± 95% confidence interval


**Table 2:** Spectral composition for the six fluorescence components identified using PARAFAC, including excitation (Ex.) and
emission (Em.) peak values, percent composition across all samples, and likely structure and characteristics of the fluorescent
component based on previous studies.

| Component | Ex. (nm) | Em. (nm) | % Composition[a] | Potential structure/Characteristics | Previous studies with comparable results |
|---|---|---|---|---|---|
| C1 | 315 | 436 | 34.1 ±2.2 (31.1-39.3) | Humic-like, less processed terrestrial, high molecular weight, widespread but highest in wetland and forest environment | Garcia et al. 2015(C1); Graeber et al. 2012(C1); Walker et al. 2014(C1); Yamashita et al. 2011(C1); Cory & McKnight, 2005(C1) |
| C2 | 270/ 380 | 484 | 20.2 ±1.9 (16.1-25.6) | Humic-like, resembles fulvic acid, widespread, high molecular weight terrestrial | Stedmon and Markager, 2005(C2); Stedmon et al. 2003(C3); Cory & McKnight, 2005(C5) |
| C3 | 270 | 478 | 17.8 ±1.8 (12.8-20.8) | Humic-like, highly processed terrestrial; suggested as refractory | Stedmon & Markager, 2005(C1); Yamashita et al. 2010(C2) |
| C4 | 305/ 435 | 522 | 14.8 ±2.6 (9.4-22.3) | Not commonly reported, similarities to fulvic-like, contributed from soils | Lochmuller & Saavedra, 1986(E) |
| C5 | 325 | 442 | 9.8 ±3.5 (0.0-15.9) | Aquatic humic-like from terrestrial environments; autochthonous, microbial produced; may be photoproduced | Boehme & Coble, 2000(Peak C); Coble et al. 1998(Peak C); Stedmon et al., 2003(C3) |
| C6 | 285 | 338 | 3.4 ±2.5 (0.0-9.3) | Amino acid-like/Tryptophan-like. Freshly added from land, autochthonous. Rapidly photodegradable | Murphy et al. 2008(C7); Shutova et al. 2003(C4); Stedmon et al. 2007(C7); Yamashita et al. 2003(C5) |

[a] Mean ± stdev (min-max) from all samples
