# Peer review of "A global hotspot for dissolved organic carbon in hypermaritime watersheds of coastal British Columbia."

_Biogeosciences, 2017_

## Referee Comment (RC1) · Anonymous Referee #1 · 15 Feb 2017

Review Oliver et al.; Globally significant yields of dissolved organic carbon from small watersheds of the Pacific coastal temperate rainforest.

General comments This manuscript examines the fluxes of DOC and the composition of DOM by stable carbon isotopes and optical measurements in several small catchments in the perhumid region of the Pacific coastal rainforest of North America. The periods of sampling for DOC concentrations and DOM composition differ but cover at least 2 years except for fluorescence measurements ($\sim$ 6 months). Analytical methods are sound. The dataset presented by the authors in such underrepresented areas represents certainly a valuable contribution for the global understanding and consideration of DOC/DOM dynamics in small rivers flowing directly into coastal waters. Overall, the

manuscript is highly (too?) descriptive and – considering the large dataset submitted by the authors – we could regret the lack of more specific goal/questions for this study. Furthermore, there are some approximation regarding DOC fluxes/yields that need to be corrected and/or clarified and I found that data were poorly included in the section aiming to describe the temporal dynamics of DOM in the catchments.

Specific comments Lines 107-110: the scientific objectives of this paper are too general.

Line 142: please define GIS.

Line 146: how have been defined the boundaries between organic and mineral soils? It would be very informative to compare a soil map and a map showing the location of wetlands and lakes within catchments.

Line 168: I suggest to use the more common notation '$\delta$13C' or '$\delta$13CDOC'.

Line 175: what size of filtration?

Line 240: define PARAFAC.

Line 301 and 306: change for 'Table 1' in the brackets.

Lines 304-310: I am not sure that is very significant. Maybe a statistical test could support this.

Line 312: why there is no DOC fluxes/yield for WY2014 while sampling for DOC has started in October 2014?

Lines 326-328: Such elevated SUVA values are commonly found in tropical rivers (e.g. Lambert et al., 2016, Biogeosciences, 13, 5405-5420) or in streams draining wetlands (e.g. Agren et al., 2008). This should be noted as it is not an exception but rather typical of environments exporting large quantities of highly aromatic DOM.

Lines 345-347: maybe this should be moved into the discussion as its interpretation of

d13C-DOC data.

Lines 325-347: where are the Fluorescence Index and the Freshness Index? Even if they have been measured over a short period, they should be described as they are included in the RDA.

Lines 348-373 & table 2: it would be informative that the corresponding number of PARAFAC components in other studies appears in table 2. For example, it's not clear what component identified in Graeber et al. (2012) matches C1. Also, according to Fellman et al. (2010), components similar to C2 are commonly reported in freshwaters. Overall this paragraph is hard to follow, mainly because figures 6 and S4.4 are not very efficient to support the text. Y-axis in figure S4.4 should be adapted for each PARAFAC component, and figure 6 could be modified in order to present temporal variability for some representative catchments. Also, some statistical tests are welcome in order to support the variability between catchments and between seasons.

Line 374: DOC export is not investigated in RDA, please change the title accordingly.

Line 374: because PARAFAC components track different fractions of the DOM pools I would suggest to perform the RDA with all components, or at least to include C3 and C5.

Line 380: I don't think that the term 'inundation' is relevant here as wetlands are wet environments. Clearly the RDA1 identifies two elements of the landscape (i.e. wetlands and lakes) as being important drivers for the spatial variability in DOC concentrations and DOM composition.

Line 384: there is no information about soil composition as only the depths of organic and mineral soil horizon have been measured.

Lines 394: the title of the section 4.1 needs to be corrected because DOC yields and DOC fluxes are calculated differently and therefore have not the same meaning. See the next comment.

Lines 395-404: this section is confusing because DOC yields and DOC fluxes have not the same meaning: DOC fluxes are the amount (mass) that passes a given point on the river over a given period of time while DOC yields are the flux per unit drainage area. If it is true that DOC yields of the study sites are higher or comparable to those estimated for some tropical rivers (higher than the Congo and the Amazon rivers but comparable relative to the Siak River), DOC fluxes are clearly lower compared to these systems (see figure). Moreover, as shown by Agren et al. (2007), DOC yields trend to decrease with catchment areas because of (1) better connection between terrestrial and aquatic ecosystems in headwater catchments, (2) reduced in stream losses in small streams and (3) increasing contribution of DOC-poor groundwater in large rivers. Consequently, the authors should compare their DOC yields to tropical catchments having similar drainage areas to support the statement that DOC yields from Calvert and Hecate Island are some of the highest recorded globally (lines 395-396). Furthermore, this statement should be taken with caution because very high DOC concentrations (> 15 mg/l) are commonly found in tropical rivers (e.g. Mayorga et al., 2003), especially in the central part of the Congo Basin where small streams < 100 km2 can have DOC concentrations up to 70 mg/l (e.g. Lambert et al., 2016, Biogeosciences, 13, 5405-5420), having thus likely among the highest DOC yields and export for streams.

Lines 408-409: is it valid for the sites studied by the authors?

Lines 405-431: this part of the manuscript is quite long and looks like more as an introduction for the section 4.2 rather than a discussion including the data.

Line 414: do you have a reference?

Lines 415-417: do you have an idea about how much represent freshwater masses compared to coastal water masses?

Lines 428-430: this phenomenon is commonly referred as 'DOC flushing' (Boyer et al., 1996, Ecological modelling, 86, 183-188) and should be moved to the beginning of section 4.2 (nothing to do with DOC fluxes/yields). Maybe the authors could also

exploit their data to discuss about hypothesis around 'DOC flushing'? Indeed, d13C-DOC values been found to investigate change in sources and pathways of DOC in small catchments (e.g. Sanderman et al., 2009, WRR, 45, W03418; Lambert et al., 2013). It is a pity that isotopic measurements made for this study are not discussed and related to the temporal and spatial variability of DOC.

Line 432: the manuscript presents no data allowing to investigate the effects of fresh DOC fluxes in coastal waters. Even if I agree that the delivery of fresh terrestrial material likely impact coastal marine foodwebs I would suggest to modify the title of the section.

Lines 433-456: it is not clear what are driving the changes in DOM composition. What are the 'microbial products and plants exudates' (line 450)? I strongly suggest to use d13C-DOC values to go deeper in this section in parallel with optical properties of DOM. How vary the fluorescence index and the freshness index and % of PARAFAC components? Is there difference in temporal variability between catchments?

Lines 457-479: ok but speculative. Maybe this could be moved at the end of the discussion. Also, potential implications of the data are included several times along the text (lines 412-414, 417-419, 431...) and consequently are quite redundant.

Section 4.3: This section can be shortened. Wetlands and lakes are known to be two major elements of the landscape having contrasting effects on DOC and DOM quality (e.g. Frost et al., 2006, Aquatic Science, 68, 40-51; Lambert et al., 2016, Biogeosciences, 13, 2727-2741) and it is relatively obvious from RDA1 that DOM concentrations and composition are largely driven by wetlands and lakes in this study. The authors should better explain the role of wetlands/lakes rather than looking for additional and questionable drivers that cannot be supported by their data.

Lines 491-492: What is the meaning of 'alternative DOC-source pools'?

Lines 491-492: watershed residence times are unknown so they can be not considered

as a driver because changes between seasons could be very low due to the small size of the catchments.

Lines 503-506: could you add figures to illustrate this?

Lines 515-516: this statement cannot be supported by the data included in the study. Soil analysis are limited to the measurements of organic and mineral horizons depths (please clarify how this has been done), and there is no information regarding soil composition (%Corg, C/N ratio, content in Al- or Fe- minerals...) that could help to investigate the role of soil composition on stream DOC dynamics. Moreover, the variability of soil organic and mineral horizons between catchments in table 1 is relatively limited. Finally, I am not convinced by the argumentation based on the RDA. For example, vectors DOC and OrgSoil have respectively negative and positive loadings along RDA1, suggesting a limited correlation. Also, the difference in the lengths of vectors C1 and Slope along RDA3 suggests that slopes are not a strong predictor for C1 (lines 529-531), as also suggested by the lack of relationship between these two vectors along RDA1.

Lines 524-526: according to recent concepts in soil science (e.g. Kaiser & Kalbitz, 2012, Soil biology & biogeochemistry, 52, 29-32), the retention of DOM in soils due to absorption processes on mineral surfaces leads to a greater biodegradation as the residence time in soil is increased. This is consistent with several studies reporting that DOC during base flow has a lower aromaticity as water pathways deepens along the soil profile (e.g. Sanderman et al., 2009, WRR, 45, W03418).

Figure 7: the variables 'OrgSoil' and 'MinSoil' are confusing between they suggest different soil composition while they are only dealing with depths. Please change their name. It is surprising that SUVA is related more to lakes rather than wetlands as the latter trend to export aromatic material. Do the authors have an explanation? DOC, d13C-DOC and C4 are clearly related to wetlands, maybe this observation should deserve more attention in the section 4.3?

Some references are missing in the reference list: Hopkinson et al., 1998; Tallis, 2009; Lambert et al., 2013

[Figure]

[Figure]

Figure – DOC fluxes and yields for tropical rivers are estimated based on published data for the Congo (Spencer et al., 2016), the Amazon (Moreira et al., 2003) and the Siak rivers (Baum et al., 2007).

**Fig. 1.**

---

## Referee Comment (RC2) · Anonymous Referee #2 · 9 Mar 2017

Review: Globally significant yields of dissolved organic carbon from small watersheds of the Pacific coastal temperate rainforest. Oliver et al.

General comments: This manuscript describes fluxes in DOC along with measurements of DOM composition (UV absorbance and fluorescence and carbon isotopes) in several catchments in the Pacific coastal temperate rainforests of North America. Overall, the data presented are interesting and important in improving our DOC flux estimates to coastal environments. A large amount of interesting data are presented however, they are not fully exploited to unpick specific research questions further than underlining the important role the catchments studied play in DOC export. I would have liked to see further analysis of the DOM compositional proxies as at present the

manuscript doesn't benefit significantly from the addition of the compositional measurements.

Specific comments:

140-142. For those not familiar with mapping software a definition of GIS would be useful. Also were catchments delineated using watershed analysis?

156-158. While less frequent sampling due to logistical constraints is understandable, have you considered how this may impact you load estimations given that large quantities of DOC that are mobilised during periods of intense rainfall? As estimates of load can be skewed significantly if large events are under represented.

218. What wavelength range did you scan over and at what interval?

219. Were high absorbing samples diluted if they breached an absorbance threshold?

228. What settings were used for your fluorescence scans (ex/em wavelengths etc.)?

240. Define PARAFAC

301. Table listed in brackets should be Table 1 not Table 2

327. The range of SUVA254 values reported in the literature is large. Elevated SUVA254 values are commonly found in both tropical rivers (Mann, P. J., et al. (2014), The biogeochemistry of carbon across a gradient of streams and rivers within the Congo Basin, J. Geophys. Res. Biogeosci., 119, 687–702, doi:10.1002/2013JG002442.) and also have been found upland peat catchment of the UK (Austnes, Kari; Evans, Chrisptoher D.; Eliot-Laize, Caroline; Naden, Pamela S.; Old, Gareth H.. 2010. Effects of storm events on mobilisation and in-stream processing of dissolved organic matter (DOM) in a Welsh peatland catchment. Biogeochemistry, 99 (1-3). 157-173. 10.1007/s10533-009-9399-4). However, lower values (<3) are also observed in groundwater dominated catchments (Yates, C, Johnes, P & Spencer, R, 2016, 'Assessing the drivers of dissolved organic matter export from two contrasting

lowland catchments, U.K'. Science of the Total Environment, vol 569-570., pp. 1330-1340).

333-347. Discussion is creeping in to the results section. Consider deleting or moving some text.

372. Could this variability be quantified in some way?

420-430. I agree with reviewer 1 one on this point. The data could be better exploited to evaluate temporal shifts in DOC/DOM composition as all the data were collected for this purpose. For example it would have been interesting if changes in DOM composition could be in some way evaluated in relation to these change in flow conditions (using either the optical measurements of 13C values). This would have given the paper more of a focus, as reviewer 1 states to investigate 'DOC flushing'.

432. Was any work done on investigating the implications of elevated DOC yields on marine foodwebs? If not then remove

490-492. What do you mean by DOC-source pools? Are you referring to the flushing of different soil horizons or the mobilising of material from a different source i.e. a source that under normal flow conditions would not be hydrologically connected to the main channel of the river? Also you have not calculated retention time for your catchments? Smaller catchments will always respond quicker than larger ones as they are simpler systems.

353. Work has already been carried out investigating long term trends in DOC flux from a wide range of catchments in relation to changes in global temperatures. See Worrall (2003). Long term records in riverine DOM. Biogeochemistry 64(2), 165-178. Or Freeman (2001) Export of organic carbon from peat soils. Nature. 412(6849) 785-785.

Figure 2. Caption is too long and bordering on discussion. Consider making more concise.

Figure 3. Are the box-whisker plots showing 1.5*IQR?

Figure 7. This also applies to the discussion but did you study catchments dominated by organic vs mineral soils or is this referring to the soil horizons? If so then consider re naming for clarity.

―――――――――――――――

---

## Referee Comment (RC3) · Anonymous Referee #3 · 11 Apr 2017

Review of Oliver et al., "Globally significant yields of dissolved organic carbon 1 from small watersheds of the Pacific coastal temperate rainforest."

This paper describes the flux and character of dissolved organic carbon (DOC) from a series of small watersheds in the Pacific coastal temperate rainforest in British Columbia. The study provides valuable new information about DOC export from a relatively understudied region. The magnitudes of the watershed DOC fluxes are impressive; however the watersheds are very small so the statements about global relevance should be tempered accordingly. The extensive dataset on DOM quality could also be better utilized to understand the mechanisms that are driving DOM export rather than just make broad observations about streamwater DOM quality. Overall, I think this

could be a solid contribution to the literature if these aspects of the paper are revised and strengthened.

General Points

There is no such thing as a "globally important" DOC yield; it is the total mass flux of DOC that could impact biogeochemical cycling on a regional or global scale. The yields reported here are quite high, however this is largely a function of the fact that DOC yields (flux per area) are inversely related to watershed size and the watersheds in this study are very small and have high wetlands coverage. Sampling a watershed that was smaller and had higher wetland coverage would produce an even higher DOC yield, however this would not make it more globally relevant. For this to represent a globally important finding, the authors would have to make the case that the fluxes measured here are broadly representative of the ∼100,000 km2 perhumid coastal forest in BC and thus provide evidence of a substantial mass flux of DOC to the coastal ocean. I understand that the purpose of this paper was not to calculate regional fluxes, however more directly addressing the issue of how regionally representative these high flux watersheds are would: 1) give readers a more concrete sense of the regional/global importance of these fluxes and 2) better justify statements such as "the small watersheds of this region export very high amounts of terrestrial DOC" (Line 477). The only place this issue is addressed in the paper is briefly in the conclusions (lines 552-554).

A similar issue arises in the discussion of the yields in section 4.1. Comparing DOC yields from these 3-10 km2 watersheds with yields from the Congo and Amazon doesn't make sense given the difference in scale. The Congo exports more than 10 Tg DOC/yr and all of the watersheds in this study together export probably 1/1000th of a Tg DOC/yr. The claim that DOC yields measured in this study are higher than those reported in southeast Asia should also be clarified given that Moore et al. (2011, 2013) have reported DOC yields >2x those reported here for watersheds in Indonesia that are several orders of magnitude larger than the watersheds in this study (doi:10.5194/bg-8-901-2011; doi:10.1038/nature11818).

There is some discussion material mixed in to the Results section of the paper. Examples include: Lines 336-339 and 345-347.

This is a very rich data set in terms of DOM compositional information. That said, the compositional data were somewhat underutilized in the study. For example, the 13C data were not even mentioned in the Discussion. In addition, the stream gage data are not utilized to elucidate how streamflow impacts DOM quality. Instead there are general statements about how compositional data change between wet and dry seasons (e.g. lines 445-456).

In Fig. 3 it appears that streamwater DOC concentrations are correlated with air temperature. If this is the case it would suggest that there is a link between soil temperature and soil water DOC production that influences the export of DOC to streams. Thus, temperature may be useful for predicting seasonal changes in streamwater DOC concentrations.

There are a number of references to watershed residence time in the Discussion (for example, lines 433, 492, 502), but it is not clear how this was quantified and whether it was function solely of lake influence or if watershed slope played a role as well.

Minor edits:

Line 74: The phrase "predictions of ecosystem productivity and food webs" is extremely vague

Lines 100-101: How and why would you expect DOC export from perhumid forests in Alaska to be different from perhumid forests in British Columbia? In other words, is there a reason to think that the work done in Alaska would not be valid in the same forest type in British Columbia?

Lines 104-105: The fact that discharge was directly measured is a strength of this study, however it is somewhat misleading to compare this highly localized study to continental and global scale studies where modeling discharge is a necessity.

[Figure]

Line 273: It seems redundant to report climatewna data in the study site and in the results. Also the values reported for mean annual precipitation differ between the study site (line 115) and the results (line 273).

Line 278: The comparison of precipitation at the study site to "most regions of the world" is vague and does not illustrate anything meaningful.

Lines 291-295: This sentence is repetitive and very hard to follow with all of the parenthetical data references. Recommend simplifying it to make the point about the difference in wet season flow without all of the Q data. It is also interesting that wet season Q differed by >20% between the two years while wet season precipitation only varied by ∼5%.

Line 326: It would be more clear to say that SUVA values were at the high end of the range rather than "relatively high compared to the range".

Line 417: "Catchment" looks like it should be plural.

Line 419: The term "a significant biogeochemical hotspot for coastal carbon cycling" is somewhat vague. Many of the studies cited in this paper calculate end of pipe DOC fluxes "directly to the coastal ocean". It would be helpful to more specifically explain why the watershed DOC fluxes in this study are "significant" from the standpoint of the coastal C cycle.

Lines 425-6: Does the term "high precipitation event" refer to intensity or magnitude. Also, it seems like the slope of these watersheds (typically >30%) is an important factor in the short hydrologic residence times that is not mentioned in this paragraph.

Lines 430-431: I agree that seasonality is important for ecological processes and it would be helpful to provide more analysis about why this would be the case in this region.

Line 455-456: Again, the consequences should be explained or this sentence should be removed.
Line 546: Because yields are a measure of the per area export (flux) of DOC the term "export the highest yields" is redundant.

---

## Author Comment (AC1) · 18 May 2017

Responses- Anonymous Referee #1

**Responses to General Comments (GC):**

Thank you for taking the time to review our manuscript and for your positive feedback on the overall scope of the paper. Below you will find our responses to your comments, which are greatly appreciated and have improved the paper. Please feel free to contact us with any additional questions or comments.

*GC1: Overall, manuscript is highly descriptive. Could regret the lack of more specific goals/questions for this study.*

Author response: The outer-central coast of British Columbia's perhumid coastal temperate rainforest is largely unstudied with respect to DOC exports, so a primary goal of this manuscript is to establish in the literature a detailed description of DOC exports for this region and put them into a global context. However, we agree that a more specific statement of goals and questions will further strengthen the quality of the manuscript. We have clarified our objectives in terms of quantifying flux and determining compositional characteristics to identify sources and drivers of DOM. We have also included some new simple statistical measures to look at differences between seasons and correlations between variables, as well as conducted a simple regional estimate of DOC flux to emphasize the importance of our results and put them into a global and regional global context.

*GC2: Some approximation regarding DOC fluxes/yields that need to be corrected and/or clarified.*

Author Response: We have attempted to clarify DOC fluxes/yields per the reviewer's comments. Please see SC19 below for complete description of our approach.

*GC3: Data poorly included in the section aiming to describe temporal dynamics of DOM.*

Author Response: We have added analysis of the role of both discharge and temperature in relation to our data and have conducted additional analysis examining differences in temporal dynamics in the dynamics of DOM compositional variables and DOC concentration. Please see SC13 and SC24 below for more information.

**Responses to Specific Comments (SC):**

*SC1: Lines 107-110: the scientific objectives of this paper are too general.*

Author Response: We have included more specific objectives/hypotheses to replace the text of lines 107-110.

*SC2: Line 142: please define GIS.*

Author Response: Text was changed to define "GIS" as "geographic information system"

*SC3: Line 146: how have been defined the boundaries between organic and mineral soils? It would be very informative to compare a soil map and a map showing the location of wetlands and lakes within catchments.*

Author Response: We have added some explanation of the distinction between organic and mineral soil materials in S1.2: "Mineral soil horizons have ≤17% organic C, while organic soil horizons have >17% organic C, as per the Canadian System of Soil Classification (Soil Classification Working Group, 1998). Boundaries between surface organic horizons and the underlying mineral soil were usually obvious, based on colour, consistence, and presence/absence of mineral grains, but for occasional ambiguous cases, grab samples were collected for laboratory determination of C content by a ThermoFischer Scientific Flash 2000 CHNS analyser at the Ministry of Environment laboratory in Victoria, B.C."

Further documentation of the soil characteristics of the watersheds will be provided elsewhere by publications in preparation, so in this manuscript we presented only the summary data needed to support the interpretations related to DOC export.

*SC4: Line 168: I suggest to use the more common notation '13C' or '13CDOC'.*

Author Response: Notation was changed to $\delta^{13}$C-DOC throughout the document.

*SC5: Line 175: what size of filtration?*

Author Response: All samples were filtered in the field using a Millipore Millex-HP Hydrophilic PES 0.45μm as described in the information on sample collection, lines 158-159.

*SC6: Line 240: define PARAFAC.*

Author Response: included definition, "..we performed parallel factor analysis (PARAFAC)…"

*SC7: Line 301 and 306: change for 'Table 1' in the brackets.*

Author Response: Changed to "Table 1" in the brackets.

*SC8: Lines 304-310: I am not sure that is very significant. Maybe a statistical test could support this.*

Author Response: We have conducted additional statistical tests that compare watersheds and seasons (wet versus dry) and that explore the relationships between DOC, stream discharge and temperature. We have reworded parts of this section to include information supported by this new analysis.

*SC9: Line 312: why there is no DOC fluxes/yield for WY2014 while sampling for DOC has started in October 2014?*

Author Response: We don't have DOC fluxes/yields for Water Year 2014 because "Water Years" are defined by the year they end (defined at the beginning of Section 3.1 "Hydrology"), and we did not have discharge data for the period of September 1, 2013 to October 1, 2014. Discharge data began October 1, 2014, and so the first year of record is Water Year 2015. This is noted in the next line, Line 313.

*SC10: Lines 326-328: Such elevated SUVA values are commonly found in tropical rivers (e.g. Lambert et al., 2016, Biogeosciences, 13, 5405-5420) or in streams draining wetlands (e.g. Agren et al., 2008). This should be noted as it is not an exception but rather typical of environments exporting large quantities of highly aromatic DOM.*

Author Response: Great point. We included a statement to clarify this, that values were "typical of environments that export large quantities of highly aromatic DOM, such as some tropical rivers (e.g., Lambert et al., 2016) or streams draining wetlands (e.g., Ågren et al., 2008)." We also added the Lambert citation to our references.

*SC11: Lines 345-347: maybe this should be moved into the discussion as its interpretation of d13C-DOC data.*

Author Response: We agree, we moved this into the discussion under Section 4.2 on "DOM Character" as an additional component aimed at better developing the interpretation of DOM sources and temporal trends/controls per the Reviewer's general comment (GC3).

*SC12: Lines 325-347: where are the Fluorescence Index and the Freshness Index? Even if they have been measured over a short period, they should be described as they are included in the RDA.*

Author Response: We included a paragraph within Section 3.3 ("Temporal and spatial patterns in DOM composition"), on the results of the Fluorescence and Freshness Index data.

*SC13: Lines 348-373 & table 2: it would be informative that the corresponding number of PARAFAC components in other studies appears in table 2. For example, it's not clear what component identified in Graeber et al. (2012) matches C1. Also, according to Fellman et al. (2010), components similar to C2 are commonly reported in freshwaters. Overall this paragraph is hard to follow, mainly because figures 6 and S4.4 are not very efficient to support the text. Y-axis in figure S4.4 should be adapted for each PARAFAC component, and figure 6 could be modified in order to present temporal variability for some representative catchments. Also, some statistical tests are welcome in order to support the variability between catchments and between seasons.*

Author Response: 1) The corresponding numbers for each of the components identified in previous studies was added to Table 2. A few references from the table were missing so they were also added to the reference list. 2) We edited the text in lines 348-373 for clarity and included some additional language to clarify that C1 and C2 are both considered to be widespread and commonly reported. 3) We adapted the Y-axis in S4.4 for each PARAFAC component and also added means and standard errors across all watersheds to give a better idea

of the spatial variability across watersheds for each component. 4) Figure 6 was modified to a panel figure in order to represent each component for all watersheds. We are hoping this addresses the reviewer's comments and better shows temporal variability for all the catchments. 5). To support the variability described between catchments and seasons, we conducted statistical comparisons to test the difference between seasons and between catchments for each component. We also looked at correlations between PARAFAC components, these are presented in a correlation matrix in the Supplementary material. These relationships are discussed in the text. A table of Pearson correlation coefficients is included as Table S4.2 in Supplementary Material.

*SC14: Line 374: DOC export is not investigated in RDA, please change the title accordingly.*

Author Response: Title changed to "Relationships between watershed characteristics, DOC concentration, and DOM composition"

*SC15: Line 374: because PARAFAC components track different fractions of the DOM pools I would suggest to perform the RDA with all components, or at least to include C3 and C5.*

Author Response: We understand the argument here and why utilizing all the components for this type (or other types) of analysis would make sense in some circumstances. Here, we are trying to identify the most important characteristics of DOM as they relate to watershed attributes. C3, C4, and C5 were removed because they were found to be highly correlated and therefore they do not appear to be significantly different from other variables as far as identifying the most important drivers of differences in bulk DOM that can be related to different watershed attributes. Because the RDA is a statistical test, leaving the correlated variables in the analysis inflates the standard errors and increases the variance of the remaining independent variables, making them more difficult to interpret in regards to teasing apart differences in watersheds and drivers of DOM/DOC concentration.

*SC16: Line 380: I don't think that the term 'inundation' is relevant here as wetlands are wet environments. Clearly the RDA1 identifies two elements of the landscape (i.e. wetlands and lakes) as being important drivers for the spatial variability in DOC concentrations and DOM composition.*

Author Response: The term 'inundation' used in this context was meant to suggest that the gradient of wetlands to lakes was a gradient of increasing water coverage (i.e., wetlands being "less inundated" to lakes being "more inundated" with water). However, it seems as though this term may be introducing confusion in this context, so we have replaced the word "inundation" with, what we hope, is a more comprehensive explanation: "a gradient of watershed coverage by inundated ecosystem types, ranging from more wetland coverage to more lake coverage".

*SC17: Line 384: there is no information about soil composition as only the depths of organic and mineral soil horizon have been measured.*

Author response: Changed "soil composition" to "soil material thickness", however an overview of the main soil types for the study area is given in lines 183-188 of the revised manuscript, as

well as in supplementary section S1.2 More details are forthcoming in publications in preparation.

*SC18: Lines 394: the title of the section 4.1 needs to be corrected because DOC yields and DOC fluxes are calculated differently and therefore have not the same meaning. See the next comment.*

Author response: This is a good point, as we do not discuss DOC flux *per se* in this section, but rather yield as the flux per unit area of watershed. Accordingly, the title of section 4.1 has been corrected to "DOC export from small catchments to the coastal ocean" as export encompasses both flux and yield.

*SC19: Lines 395-404: this section is confusing because DOC yields and DOC fluxes have not the same meaning: DOC fluxes are the amount (mass) that passes a given point on the river over a given period of time while DOC yields are the flux per unit drainage area. If it is true that DOC yields of the study sites are higher or comparable to those estimated for some tropical rivers (higher than the Congo and the Amazon rivers but comparable relative to the Siak River), DOC fluxes are clearly lower compared to these systems (see figure). Moreover, as shown by Agren et al. (2007), DOC yields trend to decrease with catchment areas because of (1) better connection between terrestrial and aquatic ecosystems in headwater catchments, (2) reduced in stream losses in small streams and (3) increasing contribution of DOC-poor groundwater in large rivers. Consequently, the authors should compare their DOC yields to tropical catchments having similar drainage areas to support the statement that DOC yields from Calvert and Hecate Island are some of the highest recorded globally (lines 395-396). Furthermore, this statement should be taken with caution because very high DOC concentrations (> 15 mg/l) are commonly found in tropical rivers (e.g. Mayorga et al., 2003), especially in the central part of the Congo Basin where small streams < 100 km2 can have DOC concentrations up to 70 mg/l (e.g. Lambert et al., 2016, Biogeosciences, 13, 5405-5420), having thus likely among the highest DOC yields and export for streams.*

Author Response: The reviewer makes some very good points. We have made the following changes:
   1) To put numbers into a more regional and global context we have included a simple regional estimate for total DOC flux from the hypermaritime region of B.C.'s perhumid coastal temperate rainforest.
   2) We included flux estimates from global to regional scales.
   3) We removed comparisons of our DOC yields with much larger rivers and instead include comparisons of watersheds of similar size, in particular those that have high amounts of precipitation, and contain extensive organic soils and wetlands. We emphasize that to the best of our knowledge, this is the first study that represents the role that these types of small catchments (high latitude, temperate, wetland and peat or organic soil-dominated) play in delivery of DOC directly to the ocean.

*SC20: Lines 408-409: is it valid for the sites studied by the authors?*

Author Response: We are emphasizing that while our sites represent small catchments, they are not first or second order headwater streams that drain to higher order catchments, but rather low to mid-order streams that drain directly to the ocean.

*SC21: Lines 405-431: this part of the manuscript is quite long and looks like more as an introduction for the section 4.2 rather than a discussion including the data.*

Author Response: This section of the manuscript was shortened and includes more discussion of the data. Lines 406-420 from the original manuscript were incorporated into the previous paragraph along with the changes discussed in the response to SC19. Lines 421-432 (from original manuscript) have been re-worded and incorporated into the beginning of Section 4.2.

*SC22: Line 414: do you have a reference?*

Author Response: Sorry, not sure of what is being referenced? Line 414 is a statement about the results of this study. We would be happy to provide a reference, or more detailed response, with clarification.

*SC23: Lines 415-417: do you have an idea about how much represent freshwater masses compared to coastal water masses?*

Author Response: It is estimated that the coastal freshwater discharge in the northeast Pacific Ocean is at least 40% of the total of freshwater that enters from the atmosphere, and is significant enough to create a freshwater-influenced water mass known as the Riverine Coastal Domain (RCD; Carmack et al. 2015). The RCD fluctuates in size but is influenced by variability in continental runoff (Morrison et al. 2012). We incorporated a sentence in the text that describes the significance of freshwater discharge and its role in the development of the RCD in this region and added the two citations mentioned above.

*SC24: Lines 428-430: this phenomenon is commonly referred as 'DOC flushing' (Boyer et al., 1996, Ecological modelling, 86, 183-188) and should be moved to the beginning of section 4.2 (nothing to do with DOC fluxes/yields). Maybe the authors could also exploit their data to discuss about hypothesis around 'DOC flushing'? Indeed, d13CDOC values been found to investigate change in sources and pathways of DOC in small catchments (e.g. Sanderman et al., 2009, WRR, 45, W03418; Lambert et al., 2013). It is a pity that isotopic measurements made for this study are not discussed and related to the temporal and spatial variability of DOC.*

Author Response: We moved the discussion of seasonal patterns in DOC to a new section ("Section 4.2: Seasonal variability in DOC export"). We also followed your suggestion to use our delta13C-DOC data, along with other measures of DOM quality and DOC concentration per comments below (SC26 re: lines 433-456), and comments from other reviewers, to further explore the relationship between discharge, temperature, DOC concentration and DOM quality. We refined our objectives to include the rationale for this additional analysis (e.g., possible seasonal and spatial trends and drivers) and to address general comments regarding incorporating

DOM data to look at temporal and cross-watershed patterns. Results are included as a figure (Figure S6.1) and two tables (Table S6.1, S6.2) in Supplementary Material.

*SC25: Line 432: the manuscript presents no data allowing to investigate the effects of fresh DOC fluxes in coastal waters. Even if I agree that the delivery of fresh terrestrial material likely impact coastal marine foodwebs I would suggest to modify the title of the section.*

Author Response: We changed the title of this section to "Sources of DOM and seasonal variability"

*SC26: Lines 433-456: it is not clear what are driving the changes in DOM composition. What are the 'microbial products and plants exudates' (line 450)? I strongly suggest to use d13C-DOC values to go deeper in this section in parallel with optical properties of DOM. How vary the fluorescence index and the freshness index and % of PARAFAC components? Is there difference in temporal variability between catchments?*

Author Response: We clarified that "microbial products and plant exudates" represent increased terrestrial primary production and microbial degradation products of lower molecular weight, less aromatic materials. Our RDA analysis looks at the role of various watershed attributes in influencing DOM composition, and in addition to box and whisker plots in Figure 2, is used to identify and discuss differences between catchments. To address questions of temporal variability, we conducted additional analysis with linear mixed effects models to see if there were relationships between DOC concentration, DOM character, and stream discharge or stream temperature. Methods and results are presented in new sections 2.7 and 3.5 "Evaluating relationships in DOC concentration and DOM composition with stream discharge and stream temperature", as well as additional discussion in section 4.3. Also see response to SC24 above for details. We also modified Figure 6 to show temporal differences between catchments for PARAFAC components. This is noted in the text under Results Section 3.3 (formerly Section 3.4).

*SC27: Lines 457-479: ok but speculative. Maybe this could be moved at the end of the discussion. Also, potential implications of the data are included several times along the text (lines 412-414, 417-419, 431. . .) and consequently are quite redundant.*

Author Response: To address the redundancy of certain points in the text related to coastal subsidies of DOC/DOM, we removed several lines of text (e.g., 417-419, 431, 478-480 (see below)). We chose to leave the text from lines 457-479 but reduced the length, because it relates to implications of patterns in the sources of DOM.

*SC28: Section 4.3: This section can be shortened. Wetlands and lakes are known to be two major elements of the landscape having contrasting effects on DOC and DOM quality (e.g. Frost et al., 2006, Aquatic Science, 68, 40-51; Lambert et al., 2016, Biogeosciences, 13, 2727-2741) and it is relatively obvious from RDA1 that DOM concentrations and composition are largely driven by wetlands and lakes in this study. The authors should better explain the role of wetlands/lakes rather than looking for additional*

*and questionable drivers that cannot be supported by their data.*

Author Response: We agree, DOM concentrations and compositions are largely driven by wetlands and lakes in this study. However, the results of our RDA also indicate other factors have significant relationships with DOM, such as depths of organic and inorganic soil types and physical watershed features such as slope. We believe it is important to include this information here, but agree that this section can be shortened. We condensed and removed much of the text on soils (lines 516-532), and text was moved to the end of the first paragraph to emphasize that the role of wetlands and underlying soils are important, but that there are also other types of non-wetland associated soils contributing sources of DOM.

*SC29: Lines 491-492: What is the meaning of 'alternative DOC-source pools'?*

Author response: We have changed the text from "DOC-source pools" to: "the contribution of DOC from sources other than organic soils associated with wetlands…"

*SC30: Lines 491-492: watershed residence times are unknown so they can be not considered as a driver because changes between seasons could be very low due to the small size of the catchments.*

Author Response: We have removed watershed residence time here as a potential driver.

*SC31: Lines 503-506: could you add figures to illustrate this?*

Author Response: This information is presented in Table 1 and in Figure 7.

*SC32: Lines 515-516: this statement cannot be supported by the data included in the study. Soil analysis are limited to the measurements of organic and mineral horizons depths (please clarify how this has been done), and there is no information regarding soil composition (%Corg, C/N ratio, content in Al- or Fe- minerals. . .) that could help to investigate the role of soil composition on stream DOC dynamics. Moreover, the variability of soil organic and mineral horizons between catchments in table 1 is relatively limited. Finally, I am not convinced by the argumentation based on the RDA. For example, vectors DOC and OrgSoil have respectively negative and positive loadings along RDA1, suggesting a limited correlation. Also, the difference in the lengths of vectors C1 and Slope along RDA3 suggests that slopes are not a strong predictor for C1 (lines 529-531), as also suggested by the lack of relationship between these two vectors along RDA1.*

Author Response: We entered additional text to clarify how organic and mineral soil depths were measured (also see response to comment SC3). We have removed most of this paragraph from the discussion (lines 516-532), including the discussion of slope as a predictor for C1 as we agree that the while slope and C1 are highly correlated along RDA3, the vector length of C1 suggests the relationship is not strong. We politely disagree with the comment on the relationship between DOC and OrgSoil, because although DOC and OrgSoil have opposite loading along Axis 1 (gradient of wetlands to lakes) they both show positive loading along Axis 2.  In an RDA

with type 2 scaling, the angle between the vectors represents the degree of correlation, with smaller angles representing more correlation. Although DOC is not the most highly correlated with OrgSoil, the angle between these vectors is <90 degrees suggesting some correlation. We changed the wording of the relationship between these variables from "important drivers of DOM composition" to "influence DOM composition" and include the caveat that this relationship is based only on depth: "However, because our study was limited to soil material depth, future work including more detailed measures of soil composition may help better describe the relationship between soils and DOM export from these watersheds."

*SC33: Lines 524-526: according to recent concepts in soil science (e.g. Kaiser & Kalbitz, 2012, Soil biology & biogeochemistry, 52, 29-32), the retention of DOM in soils due to absorption processes on mineral surfaces leads to a greater biodegradation as the residence time in soil is increased. This is consistent with several studies reporting that DOC during base flow has a lower aromaticity as water pathways deepens along the soil profile (e.g. Sanderman et al., 2009, WRR, 45, W03418).*

Author Response: This portion of text has been removed (see SC28).

*SC34: Figure 7: the variables 'OrgSoil' and 'MinSoil' are confusing between they suggest different soil composition while they are only dealing with depths. Please change their name. It is surprising that SUVA is related more to lakes rather than wetlands as the latter trend to export aromatic material. Do the authors have an explanation? DOC, d13C-DOC and C4 are clearly related to wetlands, maybe this observation should deserve more attention in the section 4.3?*

Author Response: We changed the names to "OrgThick" and "MinThick" on Figure 7.

In response to the reviewer's second comment, in a RDA of type 2 scaling, the angles between vectors of the ordination reflect their correlations, the longer a vector is along a given axis the more it contributes to that axis. With this in mind, we interpret the RDA as showing SUVA to be slightly more influenced by Axis 2 than Axis 1 (SUVA standardized coefficient score for Axis 1= 0.04, score for Axis 2= 0.07), which may represent a gradient of thicker organic to thicker mineral soil layers. However, SUVA is not strongly associated with any of the axis, suggesting the environmental variables do not explain much about our SUVA results. To help with this interpretation, we included a table (Table S5.6) of standardized coefficient scores for the DOC and DOM ("species") variables in Supplement Section 5. We also do not interpret the RDA as showing that DOC, d13C-DOC and C4 are clearly related to wetlands (also see Table S5.6). C4 is the most closely related to wetlands, but both DOC and d13C-DOC show similar or stronger correlations to OrgThick (thickness of organic soils) and Axis 2. This relates to our discussion point at the beginning of 4.3, regarding how wetlands appear not to drive the majority of observed variance in these variables. We added an additional line of text in the revised document, line 751, to restate and clarify this point.

*SC35: Some references are missing in the reference list: Hopkinson et al., 1998; Tallis, 2009; Lambert et al., 2013*

16 May 2017

Author Response: We have added those references.

---

## Author Comment (AC2) · 18 May 2017

Allison Oliver
16 May, 2017

Responses- Anonymous Referee #2

Thank you for taking the time to review our manuscript and for your positive feedback on the overall scope of the paper. Below you will find our responses to your comments, which are greatly appreciated and have improved the paper. Please feel free to contact us with any additional questions or comments.

**Responses to General Comments (GC):**

*GC1: A large amount of interesting data are presented however, they are not fully exploited to unpick specific research questions further than underlining the important role the catchments studied play in DOC export.*

Author Response: A primary goal of this manuscript is to establish in the literature a detailed description of DOC exports for this region of British Columbia and the coastal temperate rainforest and to put the results into regional and global context. However, we agree that more specific goals and questions will further strengthen the quality of the manuscript. To address this comment, we have clarified our objectives and have included additional analysis to investigate controls of flow and temperature on DOC concentration and DOM composition, a more detailed investigation of the relationship between PARAFAC components, and some simple statistical comparisons of variables across both seasons and watersheds. We also included a simple regional estimate of DOC flux to emphasize the importance of our results and put them into a global and regional global context.

*GC2: I would have liked to see further analysis of the DOM compositional proxies as at present the manuscript doesn't benefit significantly from the addition of the compositional measurements.*

Author Response: We have conducted additional analysis using linear mixed models and multiple linear regression to investigate the of discharge and temperature in relation to DOC concentration and DOM compositional data. We have also conducted additional analysis to look at relationships between PARAFAC components. Please see specific responses below for more information.

**Responses to Specific Comments (SC):**

*SC1: Line 140-142. For those not familiar with mapping software a definition of GIS would be useful. Also were catchments delineated using watershed analysis?*

Author Response: We included the definition of GIS as "geographic information system". Catchments were delineated using a 3m resolution digital elevation model (DEM) derived from airborne laser scanning (LiDAR) (Gonzalez Arriola et al., 2015). This was included in the text at the beginning of Section 2.2.

*SC2: Line 156-158. While less frequent sampling due to logistical constraints is understandable, have you considered how this may impact you load estimations given that large quantities of DOC that are mobilised during periods of intense rainfall? As estimates of load can be skewed significantly if large events are under represented.*

Allison Oliver
16 May, 2017

Author Response: We made a concerted effort to supplement our routine sampling with additional samples taken during larger events in order to better represent higher peak flows. Comparison of estimates using those additional points resulted in slightly higher load predictions for estimates that include samples from events, but no statistical comparison of the different methods has been made.

*SC3: Line 218. What wavelength range did you scan over and at what interval?*

Author Response: Included in text, "Samples were run in 1 cm quartz cells over an excitation range of 230-550 nm at 1nm increments."

*SC4: Line 219. Were high absorbing samples diluted if they breached an absorbance threshold?*

Author Response: Yes, we diluted if samples had absorbance > 0.05 at 250nm. This is included in text.

*SC5: Line 228. What settings were used for your fluorescence scans (ex/em wavelengths etc.)?*

Author Response: Included in text: "Samples were run in 1 cm quartz cells and scanned from excitation wavelengths of 230-550 nm at 5nm increments, and emission wavelengths of 210-620 nm at 2nm increments."

*SC6: Line 240. Define PARAFAC*

Author Response: Included definition in text, "parallel factor analysis"

*SC7: Line 301. Table listed in brackets should be Table 1 not Table 2*

Author Response**:** Oops, sorry! Changed to Table 1.

*SC8: Line 327. The range of SUVA254 values reported in the literature is large. Elevated SUVA254 values are commonly found in both tropical rivers (Mann, P. J., et al. (2014), The biogeochemistry of carbon across a gradient of streams and rivers within the Congo Basin, J. Geophys. Res. Biogeosci., 119, 687–702, doi:10.1002/2013JG002442.) and also have been found upland peat catchment of the UK (Austnes, Kari; Evans, Chrisptoher D.; Eliot-Laize, Caroline; Naden, Pamela S.; Old, Gareth H.. 2010. Effects of storm events on mobilisation and in-stream processing of dissolved organic matter (DOM) in a Welsh peatland catchment. Biogeochemistry, 99 (1-3). 157-173. 10.1007/s10533-009-9399-4). However, lower values (<3) are also observed in groundwater dominated catchments (Yates, C, Johnes, P & Spencer, R, 2016, 'Assessing the drivers of dissolved organic matter export from two contrasting lowland catchments, U.K'. Science of the Total Environment, vol 569-570., pp. 1330-1340).*

Author Response: We have incorporated these references into the text under the discussion in 4.2.

*SC9: Line 333-347. Discussion is creeping in to the results section. Consider deleting or moving some text.*

Author Response: We deleted some of this text, and also moved some of it to the discussion section.

*SC10: Line 372. Could this variability be quantified in some way?*

Author Response: We modified figure 6 to show results from each individual watershed, which we hope better illustrates the variability between catchments. In addition to Figure 6 and Table 1, we don't feel that it is necessary to report and compare the standard deviation associated with each sampling date as this would list the variability between different watersheds at different points in time but the figure already shows this information.

*SC11: Line 420-430. I agree with reviewer 1 one on this point. The data could be better exploited to evaluate temporal shifts in DOC/DOM composition as all the data were collected for this purpose. For example it would have been interesting if changes in DOM composition could be in some way evaluated in relation to these change in flow conditions (using either the optical measurements of 13C values). This would have given the paper more of a focus, as reviewer 1 states to investigate 'DOC flushing'.*

Author Response: We conducted additional analysis using linear mixed model multiple regression to looking at the relationships between DOC concentration and DOM compositional variables, with discharge and temperature. We refined our objectives to include the rationale for this additional analysis (e.g., possible seasonal and spatial trends and drivers) and to address general comments regarding incorporating DOM data to look at temporal and cross-watershed patterns. The methods for this additional analysis are presented in the new Section 2.7, results are presented in the new Section 3.5, and additional discussion is provided in Section 4.3. Results are also included as a figure (Figure S6.1) and two tables (Table S6.1, S6.2) in Supplementary Material.

*SC12: Line 432. Was any work done on investigating the implications of elevated DOC yields on marine foodwebs? If not then remove*

Author Response: We removed.

*SC13: Line 490-492. What do you mean by DOC-source pools? Are you referring to the flushing of different soil horizons or the mobilising of material from a different source i.e. a source that under normal flow conditions would not be hydrologically connected to the main channel of the river? Also you have not calculated retention time for your catchments? Smaller catchments will always respond quicker than larger ones as they are simpler systems.*

Author Response: By "alternate DOC-source pools" we are referring to sources of high DOC that are not associated with wetlands, typically thought of as high-DOC sources. We changed this text to: "the contribution of DOC from sources other than organic soils associated with

wetlands…" We have not calculated retention time, but based on stream flow and precipitation data we do know that catchment response time is rapid following rain events. We have provided some ancillary data in the supplementary material (Supplementary Figure S2.2) to provide a qualitative look at response time, or how quickly streams respond to precipitation. This shows, for example, the lag in response in watersheds with lakes, such as 1015 and 693. To clarify this issue, we have changed the wording "retention time" to "response time" in the text.

*SC14: Line 353. Work has already been carried out investigating long term trends in DOC flux from a wide range of catchments in relation to changes in global temperatures. See Worrall (2003). Long term records in riverine DOM. Biogeochemistry 64(2), 165-178. Or Freeman (2001) Export of organic carbon from peat soils. Nature. 412(6849) 785- 785.*

Author Response: We note this in the text and include references to previous work (including the one suggested above by Worrall).

*SC15: Figure 2. Caption is too long and bordering on discussion. Consider making more concise.*

Author Response: We made the caption more concise.

*SC16: Figure 3. Are the box-whisker plots showing 1.5*IQR?*

Author Response: Boxes represent the $25^{th}$ and $75^{th}$ percentile and whiskers represent the $5^{th}$ and $95^{th}$ percentile. We have included this in the caption for clarification.

*SC17: Figure 7. This also applies to the discussion but did you study catchments dominated by organic vs mineral soils or is this referring to the soil horizons? If so then consider renaming for clarity.*

Author Response: All watersheds contained varying areal proportions of organic (i.e. Histosols) and mineral (i.e. Podzols) soil types. The latter also contain organic horizons at the surface, of varying thickness, so the reported data for organic horizon thickness includes measurements for such cases, as well as for soils that would be classified as Histosols.

---

## Author Comment (AC3) · 18 May 2017

Responses- Anonymous Referee #3

Thank you for taking the time to review our manuscript and for your positive feedback on the overall scope of the paper. Below you will find our responses to your comments, which are greatly appreciated and have improved the paper. Please feel free to contact us with any additional questions or comments.

**Responses to General Comments (GC):**

*GC1: Statements about global relevance should be tempered accordingly because watersheds studied are so small.*

Author Response: To put our watersheds into better context, we have included additional comparisons between our watershed yields and yields from watersheds of similar size from around the world. We have also done a simple calculation of regional flux to compare potential DOC export from the region studied here, to other regions within the coastal temperate rainforest. We hope this provides better context for our flux and yield measurements relative to regional and global estimates of riverine carbon exports.

*GC2: The extensive dataset on DOM quality could also be better utilized to understand the mechanisms that are driving DOM export rather than just make broad observations about streamwater DOM quality.*

Author Response: The outer-central coast of British Columbia's perhumid coastal temperate rainforest is largely unstudied with respect to DOC exports, so a primary goal of this manuscript is to establish in the literature a detailed description of DOC exports for this region and to identify potential sources and patterns in DOM composition. We have included the RDA analysis to assess potential watershed/landscape drivers of DOC concentration and DOM composition However, agree that the manuscript benefits from more utilization of the DOM dataset. To address this comment, we have included additional analysis on the relationship between DOC and DOM composition with discharge and stream temperature. We have also done additional analysis on the relationship between PARAFAC components that goes beyond the analysis presented in the RDA. Please see specific comments below for further information.

*GC3: There is no such thing as a "globally important" DOC yield... The yields reported here are quite high, however this is largely a function of the fact that DOC yields (flux per area) are inversely related to watershed size and the watersheds in this study are very small and have high wetlands coverage. For this to represent a globally important finding, the authors would have to make the case that the fluxes measured here are broadly representative of the 100,000 km2 perhumid coastal forest in BC and thus provide evidence of a substantial mass flux of DOC to the coastal ocean. I understand that the purpose of this paper was not to calculate regional fluxes, however more directly addressing the issue of how regionally representative these high flux watersheds are would: 1) give readers a more concrete sense of the regional/global importance of these fluxes and 2) better justify statements such as "the small watersheds of this region export very high amounts of terrestrial DOC" (Line 477). The only place this issue is addressed in the paper is briefly in the conclusions (lines 552-554).*

*A similar issue arises in the discussion of the yields in section 4.1. Comparing DOC yields from these 3-10 km2 watersheds with yields from the Congo and Amazon doesn't make sense given the difference in scale. The Congo exports more than 10 Tg DOC/yr and all of the watersheds in this study together export probably 1/1000th of a Tg DOC/yr. The claim that DOC yields measured in this study are higher than those reported in southeast Asia should also be clarified given that Moore et al. (2011, 2013) have reported DOC yields >2x those reported here for watersheds in Indonesia that are several orders of magnitude larger than the watersheds in this study (doi:10.5194/bg- 8-901-2011; doi:10.1038/nature11818).*

Author Response: Thank you for this comment, you make some good points. We have made the following changes:

1) To put numbers into a more regional and global context we have included a simple regional estimate for total DOC flux from the hypermaritime region of B.C.'s perhumid coastal temperate rainforest.

2) We included flux estimates from global to regional scales.

3) We removed comparisons of our DOC yields with much larger rivers and instead include comparisons of watersheds of similar size, in particular those that have high amounts of precipitation, and contain extensive organic soils and wetlands. We emphasize that to the best of our knowledge, this is the first study that represents the role that these types of small catchments (high latitude, temperate, wetland and peat or organic soil-dominated) play in delivery of DOC directly to the ocean.

*GC4: This is a very rich data set in terms of DOM compositional information. That said, the compositional data were somewhat underutilized in the study. For example, the 13C data were not even mentioned in the Discussion. In addition, the stream gage data are not utilized to elucidate how streamflow impacts DOM quality. Instead there are general statements about how compositional data change between wet and dry seasons (e.g. lines 445-456).*
*In Fig. 3 it appears that streamwater DOC concentrations are correlated with air temperature. If this is the case it would suggest that there is a link between soil temperature and soil water DOC production that influences the export of DOC to streams. Thus, temperature may be useful for predicting seasonal changes in streamwater DOC concentrations.*

Author Response: We conducted additional analysis using linear mixed effects models to look at relationships between DOM compositional data (including 13C-DOC), DOC concentration, discharge and temperature. We refined our objectives to include the rationale for this additional analysis (e.g., possible seasonal and spatial trends and drivers) and to address general comments regarding incorporating DOM data to look at temporal and cross-watershed patterns. The methods for this additional analysis are presented in the new Section 2.7, results are presented in Sections 3.3 and 3.4, as well as included as a figure (Figure S6.1) and two tables (Table S6.1, S6.2) in Supplementary Material. Additional discussion is provided in Section 4.3.

**Responses to Specific Comments (SC):**
*SC1: There is some discussion material mixed in to the Results section of the paper. Examples include: Lines 336-339 and 345-347.*

Author Response**:** This text (and other text that bordered on discussion) has been removed from Results and is now included in Discussion. Examples given by the reviewer are now included in Section 4.3.

*SC2: There are a number of references to watershed residence time in the Discussion (for example, lines 433, 492, 502), but it is not clear how this was quantified and whether it was function solely of lake influence or if watershed slope played a role as well.*

Author Response: We did not specifically quantify residence time for watersheds, however we do know that based on the very rapid hydrograph response to precipitation events, the response time of these catchments is short. Where appropriate (such as example from line 433) we have changed this to hydrologic "response" time. In other places (such as example from 492) we removed the sentence entirely. Line 502 is providing an example from the literature, so that reference to residence times was left in the text, similar to line 512, but the text was changed here to be more explicit that this wasn't something we measured directly but an effect we would expect to see based on other watershed factors. In addition to lakes, watershed slope definitely plays a role in response/residence time, this is mentioned in line 530 of the original document and also in the results and discussion related to the RDA analysis. We have also included a figure in the Supplement (Fig. S2.2) that illustrates the response times of our watersheds with and without a high extent of lake area.

*SC3: Line 74: The phrase "predictions of ecosystem productivity and food webs" is extremely Vague*

Author Response: Changed this to just "predictions of ecosystem productivity"

*SC4: Lines 100-101: How and why would you expect DOC export from perhumid forests in Alaska to be different from perhumid forests in British Columbia? In other words, is there a reason to think that the work done in Alaska would not be valid in the same forest type in British Columbia?*

Author Response: We have included a few sentences in the text (inserted in 2[nd] to last paragraph of Introduction in new manuscript) that describe how and why we would expect DOC export to be different in the study region of B.C. versus Alaska:

"Within the large perhumid CTR, there is substantial spatial variation in climate and landscape characteristics that create uncertainty about carbon cycling and pattern. In Alaska, for example, riverine DOC concentrations vary with wetland cover (D'Amore et al. 2015) and glacial cover (Fellman et al. 2014). Previous studies have shown that streams in southeast Alaska can contain high DOC concentrations (Fellman et al., 2010; D'Amore et al., 2015a) and produce high DOC yields (D'Amore et al., 2015b; D'Amore et al., 2016, Stackpoole et al., 2016), but no known field estimates have been generated for the perhumid CTR of British Columbia, an area of approximately 97,824 km$^2$ (adapted from Wolf et al., 1995).  Within the perhumid CTR of British Columbia, terrestrial ecologists have defined a large (29,935 km$^2$) *hypermaritime* sub-region where rainfall dominates over snow, seasonality is moderated by the ocean, and wetlands are extensive (Pojar et al., 1991; area estimated using British Columbia Biogeoclimatic

Ecosystem Classification Subzone/Variant mapping Version 10, August 31, 2016, available at: https://catalogue.data.gov.bc.ca/dataset/f358a53b-ffde-4830-a325-a5a03ff672c3). Previous work in the hypermaritime CTR showed that DOC concentrations are high in small streams and tend to increase during rain events (Gibson et al., 2000; Fitzgerald et al., 2003; Emili and Price, 2013). Taken together, these conditions should be expected to generate high yields and fluxes of DOC from hypermaritime watersheds to the coastal ocean."

*SC5: Lines 104-105: The fact that discharge was directly measured is a strength of this study, however it is somewhat misleading to compare this highly localized study to continental and global scale studies where modeling discharge is a necessity.*

Author Response**:** This information was originally included to highlight the need for studies in this region that include the direct measurement of discharge, because the only work that has attempted to quantify DOC flux have been large scale studies using modeled discharge. These studies may not be appropriately capturing the heterogeneity of this complex region (see response to comment 4 above) and highlights the challenge of working in these remote locations (modelling discharge has been the only option until our paper). However, we have removed this specific text and comparison with global scale studies and now only make comparisons with regional, smaller scale studies and estimates of flux.

*SC6: Line 273: It seems redundant to report climatewna data in the study site and in the results. Also the values reported for mean annual precipitation differ between the study site (line 115) and the results (line 273).*

Author Response: We removed the second reference to climatewna in the results. Mean annual precipitation (MAP) for the study sites (line 115) is taken from sea level and central to all the study watersheds. The MAPs reported in the results (line 273 and 276) are taken from the exact location of our rain gauge and from the location of our high elevation weather station. The spatial distribution of rain in this area is extremely heterogeneous, and the range of values is presented to illustrate the differences across the landscape.

*SC7: Line 278: The comparison of precipitation at the study site to "most regions of the world" is vague and does not illustrate anything meaningful.*

Author Response: We removed "most regions of the world"

*SC8: Lines 291-295: This sentence is repetitive and very hard to follow with all of the parenthetical data references. Recommend simplifying it to make the point about the difference in wet season flow without all of the Q data. It is also interesting that wet season Q differed by >20% between the two years while wet season precipitation only varied by 5%.*

Author Response: We removed most of the parenthetical data references except for two that describe total discharge and range for water year 2015 and water year 2016. The difference in precipitation and flow between the two years is a function of, 1) precipitation arriving as snow at higher elevations that is not captured in the rain gauge, and 2) heterogeneity of rainfall across the study region. The rain gauge is centrally located in one catchment within the study region,

however this gauge probably does not capture the full range of precipitation being delivered across the islands. However, these differences would more likely be reflected in differences in Q.

*SC9: Line 326: It would be more clear to say that SUVA values were at the high end of the range rather than "relatively high compared to the range".*

Author Response: Modified the text as recommended.

*SC10: Line 417: "Catchment" looks like it should be plural.*

Author Response: This sentence has been removed.

*SC11: Line 419: The term "a significant biogeochemical hotspot for coastal carbon cycling" is somewhat vague. Many of the studies cited in this paper calculate end of pipe DOC fluxes "directly to the coastal ocean". It would be helpful to more specifically explain why the watershed DOC fluxes in this study are "significant" from the standpoint of the coastal C cycle.*

Author Response: The paragraph containing this sentence has been removed during modification of this section based on other reviewer comments.

*SC12: Lines 425-6: Does the term "high precipitation event" refer to intensity or magnitude. Also, it seems like the slope of these watersheds (typically >30%) is an important factor in the short hydrologic residence times that is not mentioned in this paragraph.*

Author Response: "High precipitation event" refers to both. We modified this sentence to reflect those details "Therefore, frequent precipitation of high magnitude or intensity …." We agree that slope is potentially an important factor influencing DOC export, and have mentioned it in Section 4.2 several times both in reference to high-gradient catchments, and the role of slope in variation between watersheds.

*SC13: Lines 430-431: I agree that seasonality is important for ecological processes and it would be helpful to provide more analysis about why this would be the case in this region.*

Author Response: These specific lines have been removed during modification of this section, however, we have incorporated text in the same area of discussion to highlight that the seasonal contribution of DOC from these watersheds to the ocean "may represent a relatively fresh, seasonally-consistent contribution of terrestrial subsidy from streams to the coastal ecosystem, which is relatively lower in carbon and nutrients throughout much of the year (Whitney et al., 2005, Johannessen et al., 2008)." The importance of seasonality in ecological processes is widely known in terms of production (both primary and secondary), and additional analysis on the importance of seasonality in terms of broader ecological processes is outside the scope of this paper.

*SC14: Line 455-456: Again, the consequences should be explained or this sentence should be removed.*

Author Response: This sentence was moved to the beginning of the following paragraph (last paragraph of Section 4.3) where we describe some of the effects of composition on biological utilization.

*SC15: Line 546: Because yields are a measure of the per area export (flux) of DOC the term "export the highest yields" is redundant.*

Author Response: Changed "export" to "contribute"

---

## Author Response (AR2)

Author response to re-review of Oliver et al., bg-2017-5, "A global hotspot for dissolved organic carbon in hypermaritime watersheds of coastal British Columbia."

Dear Associate Editor,

Please find below our response to the reviewer's comments. We appreciate all of the reviewer's insight and helpful comments, and believe we have addressed them all. However, we are happy to discuss if you would like to see something additional or different.

Thank you for your consideration and best regards,

Allison Oliver, lead author

Minor revisions: Comments Referee #1

"However, even in its revised version, I consider that the current MS is not suitable for publication and must deserve more intensive work in the interpretation of the seasonal variations and the identification of the drivers leading to seasonal changes in stream DOM concentration and composition in this part of the globe. Thus, the discussion about seasonal changes in DOM concentration and composition (sections 4.2 and 4.3, could be merged) is limited to a comparison between wet and dry periods. However, there are significant and gradual changes occurring both during wet and dry periods. For example, DOC concentrations decrease in all catchment during the wet period of the water year 2014-2015 and these decreases are linked with clear changes in DOM composition (increasing d13C-DOC, increasing SUVA and decreasing Sr). Such fluctuations contrast markedly with those reported during the previous dry period that is characterized by gradual increasing DOC concentrations, decreasing SUVA, Sr and d13C-DOC values."

*Author response*: In the last iteration of this paper, we worked hard to further investigate the relationship between seasonality and DOC/DOM beyond just "wet" vs "dry" periods. To do so, we looked at changes in DOC/DOM in relation to drivers of stream temperature and discharge as they changed throughout the year. This gave us more insight into seasonal patterns including within the predefined "wet" and "dry" periods and revealed changes in DOC/DOM parameters associated with warmer temperatures and higher discharge, both of which vary on a seasonal basis. We appreciate the reviewer's observations of seasonal patterns within the predefined "wet" and "dry" periods, and believe the results of our investigation into drivers of temperature and discharge reflect these observations. To better communicate this, and to further develop the concepts behind DOC flushing (per the reviewer's recommendation, see next comment below) we have restructured sections 4.2 and 4.3 and changed some of the wording surrounding the discussion of temperature and discharge (i.e., seasonal patterns) to avoid the misunderstanding that we are only comparing between wet and dry periods. We also specifically included the reviewer's observation that DOC concentrations increase and decrease during the dry and wet periods (revised manuscript lines 535-538) and that these changes are also associated with changes in DOM composition. We incorporated this along with the discussion of seasonallyvariable measures of stream temperature and discharge to expand the discussion of seasonality beyond the predefined periods of "wet" and "dry" (new draft lines 534-542)

The authors refer well to several studies that have investigated the DOC flushing process in other small catchments, but it has to be noted that DOC flushing is a generic term and several flushing mechanisms have been reported (Boyer et al., 1996; Sanderman et al., 2009; Lambert et al., 2013 and also Pacific et al., 2010 DOI 10.1007/s10533-009-9401-1). How the authors can better constrain DOC export in their catchments with all their dataset on DOM composition? Are these fluctuations related to (1) changes in DOM sources mobilized along the soil profile (topsoil versus subsoil, e.g. Sanderman et al., 2009), (2) changes in the production mechanisms of terrestrial DOM (e.g. Lambert et al., 2013), (3) depletion of a DOM pool depleted in 13C during the wet period (e.g. Boyer et al., 1996), (4) increasing in-stream production during the dry period…? Fluorescence data (Freshness Index, FI values and PARAFAC model) are poorly included in this part of the discussion while they can provide critical information on DOM sources and dynamic. Is there some relationships between d13C-DOC and other proxies of DOM composition that could support one of these explanations? A more robust and convincing discussion on these aspects would represent a major improvement for the MS."

*Author response*: We appreciate the reviewer's thoughtful consideration of different mechanisms related to DOC flushing and emphasis on considering these mechanisms as possible explanations for DOC/DOM export within our study. However, given the scale and scope of our study, we suggest that higher-resolution data is likely needed to specifically test these hypotheses and make conclusions about flushing mechanisms. For example, we have looked extensively at these data and expected to find relationships between DOC, d13C-DOC, and fluorescence proxies, however we were surprised not to find anything of significance. The objectives of this manuscript were to describe DOC quantity and measures of DOM composition exported from this region of the world, but not to test specific hypotheses of DOC flushing. Therefore, the next logical step for follow up to this work is to conduct studies targeted at understanding mechanisms of export. This work is currently in progress and will be included in future publications. Based on the data collected for the present study, we do not feel we are able to exclude any of the reviewer's suggested flushing mechanisms as candidates that might explain controls on export. Instead, we suggest that these possible mechanisms need to be explored further. To emphasize the reviewer's point that there are multiple possible explanations for mechanisms of DOC flushing, we have included additional points of discussion on these different mechanisms and how they may potentially apply in our study watersheds
(new manuscript lines 543-580).

Specific comments:

Lines 373-375: From my perspective, I see clear variation in d13C-DOC. Considering for example the catchment 703 (but it is also applicable for the others catchments), there is a clear and significant decrease from the dry period to the beginning of the wet period in 2014, then d13C-DOC increase along the wet period. This pattern is less obvious for the water year 2015-2016, but we can also observe a gradual increase in d13C-DOC from August 2015 to march 2016. Seasonal changes is also supported by the fact that the authors reported positive relationships between d13C-DOC, discharge and temperature (line 539). There is something here that should be investigated more deeper.

*Author response*: We would like to thank the reviewer for their detailed comment. We have changed the wording here to reflect the observation of seasonal variability/changes rather than referring to whether or not we observe a consistent and directional seasonal trend across all watersheds. In our previous revision, we made a concerted effort to further investigate drivers of seasonal patterns by looking at the specific relationship between various DOC/DOM parameters and discharge/temperature. Our results indicate that for some of our parameters, discharge and temperature, and therefore seasonal variability in these drivers, appear to be important predictors of DOC or DOM. If the reviewer (or Associate Editor) would like further analysis on seasonal changes beyond the role of discharge and temperature, we are happy to consider accommodating any suggestions that are within the scope of our dataset.

Line 482: this is partly true for the upland catchment studied by Boyer et al. (1996) (note that drier period in their catchment = winter period), but there is different "forms" of DOC flushing (See Sanderman 2009; Lambert 2013; Pacific et al., 2010 DOI 10.1007/s10533-009-9401-1).

*Author response*: To underscore the reviewer's point that there are different mechanisms and forms of DOC flushing, we have incorporated some additional text (new document lines 503-511).

Line 501: "highly terrestrial" is quite odd. "mainly terrestrial"?

*Author response*: We changed to "mainly terrestrial".

Line 519: Please clarify "wider range of DOM from soil material".

*Author response*: For clarification, we changed to "mobilization of DOM from across a wider range of the soil profile".

Line 529: Caution should be taken when interpreting relationships based on the relative contribution of PARARAC components as when one goes up, another goes down. For example, the increase in C6 suggested by the authors is not clear in Figure 7 (very low increase observed at only one date of the dry period). It will be more relevant to use maximal fluorescence intensity of components to investigate absolute increase or decrease.

*Author response*: We agree that the increased percent contribution of C6 during the dry season is largely observed during one of the sampling points. We note that the same pattern is observed in maximum fluorescence intensity (data not shown). We have removed the mention of C6 from the sentence.

Lines 539-542: these suggestions cannot be made simply based on the relationships observed between d13C-DOC and discharge and temperature. Moreover, changes in d13C-DOC are significant (> 1 ‰).

*Author response*: We removed the part of our discussion ("suggestions") highlighted by the reviewer, and included $\delta^{13}$C-DOC in the sentence below this section (updated manuscript lines 571-572) that discusses the relationship between various parameters and temperature.

Lines 542-543: Higher freshness index values do not necessary imply higher biodegradability.

*Author response*: We were not trying to connect higher Freshness Index values with higher biodegradability, rather we are suggesting that warmer temperatures lead to DOM being freshly produced, apparently from both terrestrial and microbial sources, which could potentially serve as additional source of DOM available for microbial use. To clarify this point, we have changed the language and removed the phrase "available for microbial degradation" to simply, "contribute a fresh supply of DOM exported from terrestrial sources"

Lines 546-565: this is a long discussion on an aspect of DOM cycling that has not been investigated in the study (the authors have no estimation on the biodegradability of DOM in their catchments).

*Author response*: We feel that it is important to include some preliminary discussion on the potential downstream (e.g., marine environment) implications of the high amounts of terrestrial DOC/DOM exported from these watersheds, partly to provide context for the importance of this work but also as a prelude to follow up work in preparation looking at the fate of this material in the ocean. However, we recognize the reviewer's concern about the length of a section based on information not specifically investigated in this study. In response, we have shortened this paragraph to a few sentences which are now included at the end of the previous paragraph. We also removed the citations associated with the deleted text.

Lines 578-590: is there some correlation between the relative repartition of Hemist and Folic Histosols and stream DOM concentration and composition? As data seems to be available (supplements S1.2), this would represent a strong argument to support this hypothesis.

*Author response*: While investigating specific relationships between detailed soil properties within watersheds and DOC fluxes is a next logical step, we feel it is beyond the scope of the current paper. We do, however, thank the reviewer for this insightful comment!

Lines 591-604: additional figures for illustrating the different links between lakes/wetlands and stream DOM will be great here.

*Author response:* The discussion here is related to the results of our RDA analysis, which illustrates the relationship between different DOM variables and lake and wetland area (Fig. 7). We also included an additional series of figures in the supplement (Supplemental Fig. S2.2) showing the different hydrologic responses of catchments with/without extensive lake area, as we mention lakes being important for increasing response/residence time. To aid in the assessment of ties between lake area, wetland area, and stream DOM, we have added text to the caption of Figure S2.2 to direct the reader to relevant information (Table 1) on these catchment characteristics. If the reviewer can provide more detail on a possible figure in addition to what we already provide, we would be happy to consider contributing additional information.

Reference Lambert et al. (2013) is still missing.
*Author response*: Reference now included.

[revised manuscript text omitted]

**Comment [AO2]:** Inserted per reviewer's suggestion to clarify and highlight this point.

**Comment [AO3]:** This section has been rewritten to include/outline various possible mechanisms of DOC flushing applicable in our study as suggested by the reviewer.

[revised manuscript text omitted]